# PLUG-AND-PLAY CONTROLLABLE GRAPH GENERATION WITH DIFFUSION MODELS

## ABSTRACT

Diffusion models for graph generation present transformative capabilities in generating graphs for various downstream applications. However, controlling the properties of the generated graphs remains a challenging task for these methods. Few approaches tackling this challenge focus on the ability to control for a soft differentiable property using conditional graph generation, leading to an uninterpretable control. However, in real-world applications like drug discovery, it is vital to have precise control over the generated outputs for specific features (e.g. the number of bonds in a molecule). Current diffusion models fail to support such hard non-differentiable constraints over the generated samples. To address this limitation, we propose PRODIGY (PROjected DIffusion for generating constrained Graphs), a novel plug-and-play approach to sample graphs from any pre-trained diffusion model such that they satisfy precise constraints. We formalize the problem of controllable graph generation and identify a class of constraints applicable to practical graph generation tasks. PRODIGY operates by controlling the samples at each diffusion timestep using a projection operator onto the specified constrained space. Through extensive experiments on generic and molecular graphs, we demonstrate that PRODIGY[1] enhances the ability of pre-trained diffusion models to satisfy specified hard constraints, while staying close to the data distribution. For generic graphs, it improves constraint satisfaction performance by up to 100%, and for molecular graphs, it achieves up to 60% boost under a variety of constraints.[2]

## 1 INTRODUCTION

Deep generative models serve as an effective approach to learn the underlying distribution of graph-structured data (You et al., 2018; Jo et al., 2022; Martinkus et al., 2022; Liu et al., 2019). Recently, diffusion-based models (Niu et al., 2020; Vignac et al., 2022; Jo et al., 2022; 2023) have shown impressive performance in generating graphs in an efficient manner and achieving distributional realism that outperforms most of its contemporary autoregressive and adversarial learning frameworks. The ultimate objective of the graph generation research field is to enable large-scale simulation of realistic networks that can help make tangible progress in domains such as network optimization (Xie et al., 2019), social network analysis (Grover et al., 2019), and drug design (Yang et al., 2022).

However, even with their impressive performance on benchmark datasets, current diffusion-based approaches have several limitations that keep them away from use in practice: A major limitation stems from their inability to support meaningful controllable generation. Existing methods sporadically support controllable generation (often termed conditional generation) by approximating the conditional probability distribution with the property. This approach requires the property (or its approximation) to be differentiable and it influences the sampling process in an obscure and uninterpretable manner. In real-world applications like drug discovery, precise control over the generated outputs for specific features (e.g., number of atoms in a molecule or the presence of functional groups) is crucial for a generative algorithm. Such controls are not differentiable and no method exists that can control the generation for these properties without relying on curating additional labeled datasets or retaining the entire generative model. This limitation severely restricts the applicability of these methods

---

[1]Codes have been provided as part of this submission and will be open sourced upon publication.

in graph-related applications where there are several specific structural properties that need to be controlled when generating from the pre-trained models.

In this work, we fill this gap by investigating the problem of controllable graph generation to generate graphs from an underlying distribution while satisfying certain user-defined hard constraints on its structure or properties. Specifically, we propose PRODIGY (PROjected DIffusion for generating constrained Graphs), a plug-and-play controllable generation method. Inspired by theoretical works on Projected Langevin sampling (Bubeck et al., 2018), we propose a novel sampling process that augments the denoising step with a (weighted) projection step onto the given constrained space. In this vein, we present a novel framework to devise various graph constraints and find efficient projection operators for them. Through experiments on different generic graphs, we show superior controllability of our sampling under constraints on a variety of graph properties such as edge count, triangle count, and others. PRODIGY also showcases its impressive performance in controlling molecular graph generation for constraints on a variety of properties such as valencies, atom counts, and molecular weight. This performance is further extended to 3D molecule generation, thereby demonstrating the ability of our approach to effectively handle complex structures. Finally, we conduct efficiency, sensitivity, and qualitative study to demonstrate its versatile applicability

## 2 BACKGROUND & RELATED WORK

Suppose $\mathbf{G} = (\mathbf{X}, \mathbf{A})$ denotes an undirected graph with the attribute matrix $\mathbf{X} \in \mathbb{R}^{n \times F}$ and the adjacency matrix $\mathbf{A} \in \mathbb{R}^{n \times n}$, where $n = |\mathbf{V}|$ is the number of nodes. Furthermore, a 3D structure can be defined as a point cloud $\mathbf{G} = (\mathbf{X}, \mathbf{S})$ with $\mathbf{S} \in \mathbb{R}^{n \times 3}$ denoting the positions of each node in the 3-dimensional space. Let $\mathcal{G}$ denote the set of all possible graphs (with continuous edge weights) and point clouds (for the 3D case). All vectors and matrices are represented using bold lowercase and uppercase characters. We also use $\mathbf{1}$ and $\mathbf{0}$ to denote an all-ones and an all-zeros vector with the appropriate size for the usage, e.g., in $\mathbf{A}\mathbf{1}$, $\mathbf{1}$ denotes a $n$-dimensional vector.

**Diffusion Models For Graphs.** Continuous-time diffusion models have demonstrated significant success in generating graphs for various purposes (Niu et al., 2020; Jo et al., 2022; 2023). These models are based on the idea of smoothly diffusing an unknown target distribution towards a fixed noise distribution (typically Gaussian) so that one can reverse it back to sample true data from noise. Given a graph $\mathbf{G}(0) \sim p_0$, the method follows a 'forward SDE' to gradually convert the graph into Gaussian noise, *i.e.*, $\mathbf{G}(T) \sim p_T = \mathcal{N}(\boldsymbol{\mu}, \boldsymbol{\Sigma})$, for a fixed $\boldsymbol{\mu}, \boldsymbol{\Sigma}$.

$$d\mathbf{G} = \mathbf{f}(\mathbf{G}, t)dt + g(t)d\mathbf{w}, \tag{1}$$

where $\mathbf{f} : \mathcal{G} \times t \to \mathcal{G}$ is the *drift coefficient*, $g : \mathbb{R} \to \mathbb{R}$ is the *diffusion coefficient*, and $\mathbf{w}(t) \in \mathcal{G}$ is a standard Wiener process. In order to generate samples from the unknown data distribution $p_0$, the forward process is reversed so that samples from the prior distribution can be converted to the target distribution. Anderson (1982) shows that the reverse process can be given as:

$$d\mathbf{G} = [\mathbf{f}(\mathbf{G}, t) - g(t)^2 \nabla_{\mathbf{G}} \log p_t(\mathbf{G})]dt + g(t)d\bar{\mathbf{w}}, \tag{2}$$

where $\bar{\mathbf{w}}$ is a reverse-time standard Wiener process and $p_t$ denotes the marginal distribution of $\mathbf{G}_t$ at time-step $t$. $\nabla_{\mathbf{G}} \log p_t(\mathbf{G})$ is called the score function at time $t$.

Since the time-conditional score function is not available for an unknown distribution $p_0$, one estimates it with a parameterized neural network $\mathbf{s}_\theta(\mathbf{G}, t) \approx \nabla_{\mathbf{G}} \log p_t(\mathbf{G})$ by minimizing a score-matching objective across multiple time steps and training samples. EDP-GNN (Niu et al., 2020) ignores the diffusion process of $\mathbf{X}$ and samples directly from the prior distribution of $\mathbf{X}$. GDSS (Jo et al., 2022) considers a system of SDEs to efficiently estimate score functions for $\mathbf{X}$ and $\mathbf{A}$ separately. DruM (Jo et al., 2023) models the graph topology by conditioning the process on the destination data distribution using a mixture of Ornstein-Uhlenbeck processes. These models have also been proposed to predict structures in the 3-dimensional space by generating the positions $\mathbf{S}$ and types of each node in the 3D space (Hoogeboom et al., 2022; Xu et al., 2023).

An alternative set of approaches (Vignac et al., 2022; Chen et al., 2023) focus on extending discrete diffusion models (Austin et al., 2021) for the graph generation. Specifically, these models sample a discrete graph $\mathbf{G}_t$ from a noisy probability distribution at each timestep $t$. These approaches attempt to alleviate the following drawbacks that continuous-time models may encounter when performing graph generation: the destruction of graph sparsity and extension to arbitrarily many atom types.

However, in doing so, they lose the ability to facilitate *interpretable control* during generation due to the combinatorial explosion of the discrete space that satisfies a constraint and the lack of the gradient information, that is available in the continuous space. Further, most recent advancements in continuous-time models (Jo et al., 2023) have been shown to significantly outperform existing discrete approaches. For these reasons, this work focuses on providing interpretable control for continuous-time diffusion models and leaves its extension to discrete approaches for future studies.

**Controlled Generation.** In real-world settings, it is often crucial for practitioners to have control over the properties of the generated graph, for instance, constraining the number of atoms of a certain type or the total molecular weight. This is an open and challenging problem with two possible types of controlled generation: (i) Condition-based Control (Soft) and (ii) Constraint-based Control (Hard). Below we discuss the properties and usefulness of each type through existing works.

*1. Condition-based Control (Soft).* The generated outputs are controlled by approximating a conditional probability distribution $p(\mathbf{G}|c) := p(\mathbf{G}|c(\mathbf{G}, y))$. Typically, the condition $c(\mathbf{G}, y) = \mathbb{1}\{y_c(\mathbf{G}) = y\}$, i.e., $c(\mathbf{G}, y) = 1$ if $y_c(\mathbf{G}) = y$ and 0 otherwise. Note that this does not guarantee that $c(\mathbf{G}, y)$ will hold true for the generated output $\mathbf{G}$. To the best of our knowledge, there have been sporadic attempts to support control with diffusion models of graph generation, and all existing works in this space fall under this category of soft control. Conditional denoising models (Hoogeboom et al., 2022; Xu et al., 2023) learn a conditional score function $s_\theta(\mathbf{G}, c) \approx \nabla \log p(\mathbf{G}|c)$. Thus, each condition type demands a unique model and cannot be used in a plug-and-play manner for an unconditional model as it requires retraining the model for a new control. On the other hand, guidance-based diffusion methods (Vignac et al., 2022; Graikos et al., 2022; Lee et al., 2023; Li et al., 2022) infer $p(\mathbf{G}|c)$ from $p(c|\mathbf{G})$ by using the fact that $\nabla \log p(\mathbf{G}|c) \approx s_\theta(\mathbf{G}) + \nabla \log p(c|\mathbf{G})$. This allows for plug-and-play conditional control on pre-trained diffusion models $s_\theta(\mathbf{G})$ as long as we can approximate $\nabla \log p(c|\mathbf{G})$. When $c(\cdot)$ is not known, it is approximated by a classifier $\hat{y}_c$ while when it is known, the property $c$ is assumed to be a differentiable function of $\mathbf{G}$ and $y$. Classifier-based guidance requires labeled data along with capabilities to train a classifier for every new control.

Thus, it is impossible to directly apply condition-based control methods to our setting where we want plug-and-play control with constraint functions that are known but not differentiable.

*2. Constraint-based Control (Hard).* Precise control on the generated output can be formulated in terms of specific well-defined constraints. For example, if $\mathcal{C}$ is a user-defined constraint set, then we have $c(\mathbf{G}) = 1$ if $\mathbf{G} \in \mathcal{C}$ and 0 otherwise. We note that this identity function $c(\mathbf{G})$ is non-differentiable. Bar-Tal et al. (2023) recently proposed preliminary steps in this direction with a focus on the task of image generation. This approach supports specific image constraints such as panorama, aspect ratio, and spatial guiding, by solving an optimization problem to match the pre-trained sampling process in the constrained space. To the best of our knowledge, no prior work exists that can support the constrained-based generation of graphs using diffusion models.

The main contribution of our work is to provide an efficient and general (supports arbitrary hard constraints) method grounded in the projected sampling process to enable constrained generation for graphs using diffusion models in a plug-and-play manner.

**Projected Sampling.** In the literature, the theoretical ability of projected/mirrored Langevin sampling to enable constrained sampling from underlying distribution has been explored (Bubeck et al., 2018; Hsieh et al., 2018). However, its effectiveness for deep learning-based diffusion models is still unknown as the underlying distribution is approximated from the training data, which would render sampling infeasible in uncertain domains. Furthermore, diffusion models employ additional reverse-SDE dynamics on top of the simple Langevin sampling. Finally, efficient projections for many graph-level constraints need to be derived for application in this framework. In this work, we address all these challenges by studying newly proposed variants of projected sampling in the realm of modern diffusion models under a novel set of graph constraints.

## 3   PROBLEM SETUP: PLUG-AND-PLAY CONSTRAINED GRAPH GENERATION

Given a set of training graphs $\mathcal{G}_{tr} \subset \mathcal{G}$, the problem of graph generation involves learning the underlying distribution $p_0$ as $\mathcal{G}_{tr}$ and sampling from the learned distribution to generate new graphs $\{\mathbf{G}\}$ such that they mimic the training distribution $p_0$. In this work, we consider the problem of constrained graph generation, where the objective is to control the generative process within a given constrained set. Specifically, we solve the following problem:

**Problem 1.** *(Plug-and-Play Constrained Graph Generation) Given a constrained set $\mathcal{C} \subseteq \mathcal{G}$ and a pre-trained unconditional graph generation model $\mathcal{M}$ trained on some training set $\mathcal{G}_{tr} \sim p_0$, generate new graphs $\{\mathbf{G}\} \sim \hat{p}_0$ such that $\hat{p}_0 \approx p_0$ and $\hat{p}_0$ has support $\mathcal{C}$.*

**Key Assumption:** The model $\mathcal{M}$ may not be available for further training nor do we have access to training set $\mathcal{G}_{tr}$ or the model parameters $\mathbf{\Theta}(\mathcal{M})$ as these are often not released due to proprietary reasons (Ramesh et al., 2021; OpenAI, 2023). Thus, the proposed method is required to be flexible to the choice of the the constraints and the underlying generative model (plug-and-play approach). Next, we discuss the class of constraints we study in this work.

### 3.1 CONSTRAINTS

For this work, we consider a wide range of arbitrary constraints with focus on interpretability and minimal restrictions. Concretely, our approach is able to handle any constraint of the form $\mathcal{C} = \{\mathbf{G} : h_1(\mathbf{G}) \leq 0, h_2(\mathbf{G}) \leq 0, \cdots, h_k(\mathbf{G}) \leq 0\}$, with efficient solutions of simultaneous equality. As such, a practitioner may be interested in controlling the generation with a variety of constraints on the structure and the derived properties of the graph, depending on the downstream applications. To this end, we motivate our approach by instantiating a set of constraints based on well-studied graph properties in both generic graph structures (with applications to network design and efficiency) and molecules (with applications to drug design). Below we discuss the key set of constraints that we instantiate to provide the recipe of our approach and further discuss extensions towards more complex properties in Appendix B.

**Generic Graphs.** A user may want to control the number of different substructures in the graph (Tabourier et al., 2011; Ying and Wu, 2009) since these represent different aspects of real-world network design (Farahani et al., 2013). We consider three such constraints (adjoining table) since Edge Count reflects a budget on the total (unweighted) cost of roads, while degree and triangle count measure network efficiency for the load on a node, and local clustering respectively.

| | |
|---|---|
| **Edge Count** | Number of edges $\|\mathbf{E}\| = \frac{1}{2}\mathbf{1}^T\mathbf{A}\mathbf{1}$ $\leq \mathcal{B}$ for a given constant $\mathcal{B} \geq 0$ |
| **Triangle Count** | Number of triangles $\frac{1}{6}\mathrm{tr}(\mathbf{A})$ $\leq T$ for a given constant $T \geq 0$ |
| **Degree** | Degree of each node is bounded by a constant, *i.e.*, $\mathbf{A}\mathbf{1} \leq \delta_d\mathbf{1}$ |

**Molecular graphs.** In the adjoining table, we have $\mathbf{X}$ denoting one-hot encoding of each node being a certain atom $\in \{1, 2, \cdots, F\}$. It is often desired that the generated molecule is valid (Vignac et al., 2022; Jo et al., 2022) and has some desired properties. Chemical descriptors (Todeschini and Consonni, 2008) link molecular structure to its properties. At a base level, a molecular structure is comprised of atoms $\mathbf{X}$, their connections $\mathbf{A}$ and 3D positions $\mathbf{S} \in \mathbb{R}^{n \times 3}$.

| | |
|---|---|
| **Valency** | Given valencies $\mathbf{v}$, degree is at most valency[3], *i.e.*, $\mathbf{A}\mathbf{1} \leq \mathbf{X}\mathbf{v}$ |
| **Atom Count** | Number of atoms of each type is bounded, *i.e.*, $\mathbf{X}^T\mathbf{1} \leq \mathbf{c}$, for counts $\mathbf{c}$ |
| **Molecular Weight** | Total weight is bounded, *i.e.*, $\mathbf{1}^T\mathbf{X}\mathbf{m} \leq W$, for atomic weights $\mathbf{m}$ |
| **Dipole Moment** | Norm of the vector sum of the atomic charges is bounded, *i.e.*, $\|\mathbf{S}^T\mathbf{X}\mathbf{Q}\|_2 \in [\xi_0, \xi_1]$ |

## 4 PROPOSED METHOD: PROJECTED DIFFUSION FOR CONSTRAINED GRAPHS

We propose PROjected DIffusion for constrained Graphs (PRODIGY), a plug-and-play sampling method for constrained graph generation for continuous-time diffusion models. Figure 1 illustrates our method and how it enables an arbitrary constraint to be satisfied. Following theoretical works in Mirrored Langevin Dynamics (Bubeck et al., 2018; Hsieh et al., 2018), we extend the idea of Projected Gradient Descent to constrained sampling by alternate dynamics and projection.

$$\begin{cases} \widetilde{\mathbf{G}}_{t-1} \leftarrow \text{Reverse}(\mathbf{G}_t, \bar{\mathbf{w}}_t, t; \mathbf{f}, g, \mathbf{s}_\theta) \\ \mathbf{G}_{t-1} \leftarrow \Pi_{\mathcal{C}}(\widetilde{\mathbf{G}}_{t-1}), \end{cases} \tag{3}$$

where Reverse is some arbitrary discretization of the reverse process defined in Equation 2 with score function $\mathbf{s}_\theta$ and $\Pi_{\mathcal{C}}$ is the projection operator, $\Pi_{\mathcal{C}}(\mathbf{x}) = \arg\min_{\mathbf{z} \in \mathcal{C}} \|\mathbf{z} - \mathbf{x}\|_2^2$. Figure 2 illustrates

---

[3]We assume hidden Hydrogen atoms, following Jo et al. (2022); Vignac et al. (2022).

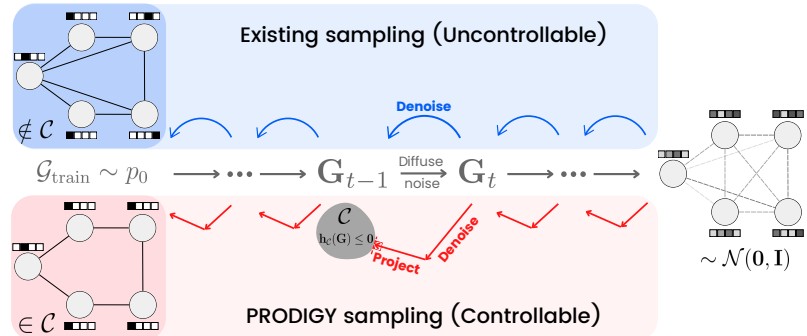

Figure 1: Comparison of existing and proposed projected diffusion sampling methods for generating graphs under the given constrained set $\mathcal{C}$ (*e.g.*, number of edges is at most 5).

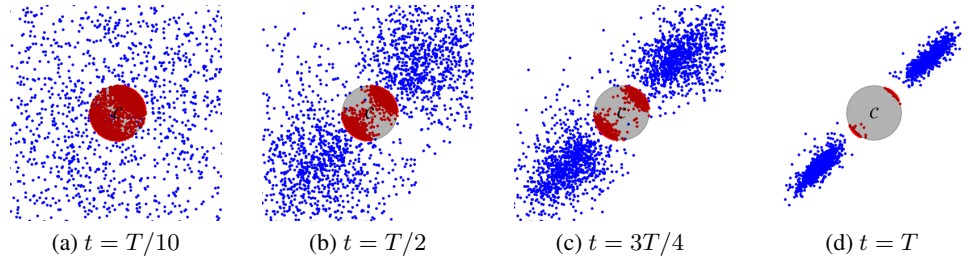

| (a) $t = T/10$ | (b) $t = T/2$ | (c) $t = 3T/4$ | (d) $t = T$ |

Figure 2: Sampling process of PRODIGY (red) versus existing methods (Jo et al., 2022) (blue) at different diffusion timesteps ($t$) for a constrained generation within an $\ell_2$ ball centered at the origin and a radius of $0.1$. PRODIGY generates points at the periphery of the constrained set $\mathcal{C}$, closest to the data density. The original distribution is a Gaussian mixture model of two equally likely normal distributions with means $(1, 1)$ and $(-1, -1)$ and symmetric covariances of $0.01$ and $0.009$.

the sampling process of our method as compared to the existing sampling strategies. PRODIGY is able to sample within the constrained $\ell_2$ ball while existing strategies fail to do that.

However, we also note that projection to an arbitrary constrained set can destroy the smoothness of the reverse process. This is because the denoised sample can exist very far from the feasible set and the projection step might go into a region with low estimated probability at a timestep $t$. To account for this, we propose to take only a $\gamma_t$ step towards the constrained set from $\widetilde{\mathbf{G}}_{t-1}$. In particular, we consider $\mathbf{G}_{t-1} \leftarrow (1 - \gamma_t) \widetilde{\mathbf{G}}_{t-1} + \gamma_t \Pi_{\mathcal{C}}(\widetilde{\mathbf{G}}_{t-1})$. One can note that a higher $\gamma_t$ implies a higher chance of constraint satisfaction but also a higher distortion to the original sampling process. Thus, we should select a lower $\gamma_t$ when the focus is to approximate the underlying distribution and a higher $\gamma_t$ to satisfy the constraint. We handle this tradeoff by considering a polynomial schedule w.r.t. the diffusion timestep. Initially, when $t \approx T$, approximating the actual distribution is more important, thus, $\gamma_t \approx 0$. As $t \to 0$, the graphs are close to the original distribution and we should focus on constraint satisfaction (thus, $\gamma_t \to 1$). Therefore, we consider a smooth polynomial function: $\gamma_t = \text{poly}(\gamma_T, t) = (1 - \gamma_T)\left(\frac{T-t}{T}\right)^r + \gamma_T$, for some $\gamma_T, r \geq 0$. Next, we discuss the projection operators $\Pi_{\mathcal{C}}(\mathbf{G})$ to transform a given graph ($\mathbf{G}$) to its closest counterpart that satisfies a given constraint from the set of constraints discussed in Section 3.1.

### 4.1 PROJECTION OPERATORS

Consider a constraint of the form $\mathcal{C} = \{\mathbf{Z} = (\mathbf{Z}_X, \mathbf{Z}_A) \in \mathcal{G} : \mathbf{h}_{\mathcal{C}}(\mathbf{Z}) \leq \mathbf{0}\}$ on the set of graphs. Then, the projection operator is given as:

$$\Pi_{\mathcal{C}}(\mathbf{G}) = \underset{\substack{(\mathbf{Z}_X, \mathbf{Z}_A) \in \mathcal{G} : \mathbf{h}_{\mathcal{C}}(\mathbf{Z}_X, \mathbf{Z}_A) \leq \mathbf{0} \\ \mathbf{Z}_X \in [\mathbf{X}_m, \mathbf{X}_M], \mathbf{Z}_A \in [\mathbf{A}_m, \mathbf{A}_M] \\ \mathbf{Z}_A^T = \mathbf{Z}_A, \text{Diag}(\mathbf{Z}_A) = \mathbf{0}}}{\arg\min} \frac{1}{2}\|\mathbf{Z}_X - \mathbf{X}\|_2^2 + \frac{1}{2}\|\mathbf{Z}_A - \mathbf{A}\|_2^2, \quad (4)$$

Table 1: Projection Operators for different constraints, given as $\Pi_{\mathcal{C}}(\mathbf{G}) = \varphi_{\mathbf{0}}(\mathbf{G})$ if $\mathbf{h}_{\mathcal{C}}(\varphi_{\mathbf{0}}(\mathbf{G})) \leq \mathbf{0}$ otherwise $\varphi_{\boldsymbol{\mu}}(\mathbf{G})$ such that $\mathbf{h}_{\mathcal{C}}(\varphi_{\boldsymbol{\mu}}(\mathbf{G})) = \mathbf{0}$. See App A for proofs and extensions.

| 2D structure $\mathbf{G} = (\mathbf{X}, \mathbf{A})$. $\mathbf{X} \in [0,1]$, $\mathbf{A} \in [0,1]$ or $\in [0,3]$, $\mathbf{A}^T = \mathbf{A}$, $\text{Diag}(\mathbf{A}) = \mathbf{0}$ | | | |
|---|---|---|---|
| Constraint ($\mathcal{C}$) | Function ($\mathbf{h}_{\mathcal{C}}$) | $\varphi_{\boldsymbol{\mu}}^{X}$ | $\varphi_{\boldsymbol{\mu}}^{A}$ |
| Edge Count | $\frac{1}{2}\mathbf{1}^T\mathbf{A}\mathbf{1} - \mathcal{B}$ | $\mathbf{X}$ | $P_{[0,1]}(\mathbf{A} - \mu\mathbf{1}\mathbf{1}^T/2 + \mathbf{I}/2)$ |
| Triangle Count | $\frac{1}{6}\text{tr}(\mathbf{A}^3) - T$ | $\mathbf{X}$ | $P_{[0,1]}(\mathbf{A} - \mu\mathbf{A}^2/2)$ |
| Degree | $\mathbf{A}\mathbf{1} - \delta_d\mathbf{1}$ | $\mathbf{X}$ | $P_{[0,1]}(\mathbf{A} - \frac{1}{2}(\boldsymbol{\mu}\mathbf{1}^T + \mathbf{1}\boldsymbol{\mu}^T) + \text{Diag}(\boldsymbol{\mu}))$ |
| Valency | $\mathbf{A}\mathbf{1} - \mathbf{X}\mathbf{v}$ | $P_{[0,1]}(\mathbf{X})$ | $P_{[0,3]}(\mathbf{A} - \frac{1}{2}(\boldsymbol{\mu}\mathbf{1}^T + \mathbf{1}\boldsymbol{\mu}^T) + \text{Diag}(\boldsymbol{\mu}))$ |
| Atom Count | $\mathbf{X}^T\mathbf{1} - \mathbf{c}$ | $P_{[0,1]}(\mathbf{X} - \mathbf{1}\boldsymbol{\mu}^T)$ | $P_{[0,3]}(\mathbf{A})$ |
| Molecular Weight | $\mathbf{1}^T\mathbf{X}\mathbf{m} - W$ | $P_{[0,1]}(\mathbf{X} - \mu\mathbf{1}\mathbf{m}^T)$ | $P_{[0,3]}(\mathbf{A})$ |

| 3D structure $\mathbf{G} = (\mathbf{X}, \mathbf{S})$. Attributes $\mathbf{X} \in [0,1]$, Positions $\mathbf{S} \in \mathbb{R}^{n\times 3}$ | | |
|---|---|---|
| Constraint ($\mathcal{C}$) | Function ($\mathbf{h}_{\mathcal{C}}$) | $\varphi_{\boldsymbol{\mu}}^{X}$ $\quad\quad\quad$ $\varphi_{\boldsymbol{\mu}}^{S}$ |
| Dipole Moment | $\xi_0 \leq \|\mathbf{S}^T\mathbf{X}\mathbf{Q}\|_2 \leq \xi_1$ | $\mathbf{X}$ $\quad\quad\quad$ $\mu\mathbf{S}/\|\mathbf{S}^T\mathbf{X}\mathbf{Q}\|_2$ |

which can be solved using the Lagrangian method, $\mathcal{L}(\mathbf{Z}_X, \mathbf{Z}_A, \mathbf{h}_{\mathcal{C}}, \boldsymbol{\lambda}, \boldsymbol{\mu}) = \frac{1}{2}\|\mathbf{Z}_X - \mathbf{X}\|_2^2 + \frac{1}{2}\|\mathbf{Z}_A - \mathbf{A}\|_2^2 + \boldsymbol{\mu}_0 \cdot \mathbf{h}_{\mathcal{C}}(\mathbf{Z}_X, \mathbf{Z}_A) + \boldsymbol{\mu}_1 \cdot (\mathbf{Z}_X - \mathbf{X}_m) + \boldsymbol{\mu}_2 \cdot (\mathbf{X}_M - \mathbf{Z}_X) + \boldsymbol{\mu}_3 \cdot (\mathbf{Z}_A - \mathbf{A}_m) + \boldsymbol{\mu}_4 \cdot (\mathbf{A}_M - \mathbf{Z}_A) + \sum_{i>j} \lambda_{ij}(\mathbf{Z}_A[i,j] - \mathbf{Z}_A[j,i]) + \sum_i \lambda_i \mathbf{Z}_A[i]$. We apply the Karush–Kuhn–Tucker (KKT) conditions (Kuhn and Tucker, 2013) and find closed-form solutions for $\mathbf{Z}_X$ and $\mathbf{Z}_A$. For 3D structures, we consider the positions $\mathbf{S}$ instead of $\mathbf{A}$ with no additional constraints on $\mathbf{Z}_S$.

Table 1 lists the projection operators for different constraint functions. Please refer to Appendix A for the complete derivations for each case. We note that for several constraints, $\mathbf{h}_{\mathcal{C}}$ and $\boldsymbol{\mu}$ are scalars. Thus, we solve for $\mu$ in $h_{\mathcal{C}}(\varphi_\mu(\mathbf{G})) = 0$ using the bisection method (Boyd et al., 2004). When $\mathbf{h}_{\mathcal{C}}$ (and thus, $\boldsymbol{\mu}$) are vectors (as in the Degree, Valency, and Atom Count constraints), we split $\mathbf{h}_{\mathcal{C}}$ into independent functions $h_{\mathcal{C}}^{(i)}$ and solve for $\mu_i$ such that $h_{\mathcal{C}}^{(i)}(\varphi_{\mu_i}(\mathbf{G})) = 0$ using the bisection method. The split is done such that if $h_{\mathcal{C}}^{(i)}(\varphi_{\mu_i}(\mathbf{G})) = 0$ for all $i \in [1, M]$, then for $\boldsymbol{\mu} = (\mu_1, \mu_2, \cdots, \mu_M)$, $\mathbf{h}_{\mathcal{C}}(\varphi_{\boldsymbol{\mu}}(\mathbf{G})) \leq \mathbf{0}$. Thus, the obtained solution would satisfy the constraint. This sampling approach is highly efficient, which we discuss in detail in Appendix D.2.

## 5 EXPERIMENTS

To test the efficacy of our method, we ask and investigate the following questions — **(1)** Can PRODIGY effectively generate graphs that satisfy hard constraints on their structure and properties? **(2)** Can PRODIGY effectively handle more complex structures such as 3D molecules? **(3)** How does the PRODIGY sampling process affect the distributional properties learned by the original model? **(4)** What is the sensitivity of our approach to tuning parameters and constraint constants? **(5)** How efficient is PRODIGY and how does the generated graphs fare qualitatively? We address the first two questions in positive by quantifying the generation performance under set of constraints on both generic and molecular graphs (Sec. 5.2, Sec. 5.3). Next, in Sec. 5.4, we consider a setting that would mimic unconstrained setting while actually satisfying the specified constraint, thereby facilitating to quantify how well the PRODIGY approach preserves the distribution learned by underlying model. Sec. 5.5 contains the sensitivity analysis and we report run time analysis and qualitative visualization of the generated graphs in the Appendices D.2 and D.3 respectively.

### 5.1 SETUP

Before diving into experimental results, we briefly outline the important details on datasets, constraints, baselines and metrics in this section and provide more elaborate details in Appendix C.

**Datasets.** We consider four generic and two molecular datasets to evaluate the ability of PRODIGY to generate good-quality constrained graphs. Generic graph datasets include Community-small, Ego-small, Grid (You et al., 2018), and Enzymes (Jo et al., 2022). In addition, we also consider two standard molecular datasets: QM9 (Ramakrishnan et al., 2014) and ZINC250k (Irwin et al., 2012). For a fair comparison, we follow the standard experimental setup of existing works (You et al., 2018; Jo et al., 2022; Niu et al., 2020).

**Constraints.** As noted in Section 3.1, we consider the constraints of edge count, triangle count, and degree for non-attributed generic graphs. On the other hand, we evaluate the controllability in attributed molecular graph generation under valency, atom count, and molecular weight constraints. Each constraint consists of an extrinsic control parameter that we vary to assess the effectiveness of our approach for a range of values.

**Base models.** To evaluate the plug-and-play ability of our method on 2D graph generation, we evaluate its performance on two baseline continuous-time diffusion models[4]: (1) EDP-GNN (Niu et al., 2020), and (2) GDSS (Jo et al., 2022). For 3D graph generation, we use the pre-trained model of GeoLDM for QM9 (Xu et al., 2023).

**Metrics.** We assess the performance of our method towards satisfying the given constraint and also report various distributional metrics. For the former, we consider the proportion of generated graphs that satisfy the constraint, i.e., $\text{VAL}_\mathcal{C}(\mathbf{G}) := \frac{1}{N} \sum_{i \in N} \mathbb{1}[\mathbf{G}_i \in \mathcal{C}]$, where we generate $N$ different graphs $\{\mathbf{G}_i\}$. To evaluate the distributional preservation under our approach, we compared the distributions of certain graph statistics between generated and test graphs using the maximum mean discrepancy (MMD) metric (You et al., 2018). For molecules, we use the Fréchet ChemNet Distance (FCD) (Preuer et al., 2018) and Neighborhood Subgraph Pairwise Distance Kernel (NSPDK) (Costa and De Grave, 2010). In addition, we also consider the validity, uniqueness, and novelty metrics.

## 5.2 CAN PRODIGY EFFECTIVELY GENERATE GRAPHS THAT SATISFY HARD CONSTRAINTS?

**Generic Graphs.** We consider a constraint that is satisfied by the least number of graphs in the test set and generate graphs that satisfy it. For example, for a property $\mathcal{P}(G)$, we consider the property of the generated graph $\mathcal{P}(\mathbf{G})$ to follow $\mathcal{P}(\mathbf{G}) \leq \min_{G \in \mathcal{G}_{ts}}\{\mathcal{P}(G)\}$. We compare the MMDs and the constraint validity ($\text{VAL}_\mathcal{C}$) of the generated generic graphs for each of the three constraints on generic graphs. Table 2 shows the effect of plugging PRODIGY for sampling the two base models under these constraints for different datasets. We can note that the constraint validity with PRODIGY sampling is almost always close to $1$, *i.e.*, almost all the generated graphs satisfy the constraint, with the minimum being $65\%$ for GDSS in the Ego-small dataset for the Edge Count constraint. PRODIGY increases the constraint validity in GDSS by at least $20\%$ and at most $100\%$ across 4 datasets and 3 different constraints. We also find that the MMDs between the generated graphs and the constraint-filtered graphs under PRODIGY sampling are similar to the original sampling, with improvements in some cases. One exception is Grid, where the MMD scores increase under this constrained setting. We do more analysis of these cases in Appendix D.3 and D.5.

**2D Molecular Graphs.** Table 3 shows the effect of plugging PRODIGY to generate molecules with specified properties. For the valency constraint, we consider the valencies $C_4N_5O_2F_1$ in QM9 and $C_4N_3O_2F_1P_5S_2Cl_1Br_1I_1$ in ZINC250k. For the Atom Count, we constrained the generated molecule to only contain C and O for both QM9 and ZINC250k. Lastly, we constrain the molecular weight of the generated molecules to be within the lower 10-percentile range of the test set. We find that PRODIGY always improves (or matches) the constraint validity across the two datasets, while not compromising on the other metrics. PRODIGY improves the constraint validity by at least $1\%$ and up to $60\%$ for GDSS, while also improving FCD and NSPDK from the constraint-filtered test graphs by up to $12.92$ and $0.13$ points respectively. We also see similar gains in performance for EDP-GNN.

## 5.3 CAN PRODIGY CONSTRAIN 3D MOLECULE GENERATION?

Here, we show the effect of using PRODIGY sampling on 3D molecule generation. We use the dipole moment (as formulated in Sections 3.1, 4.1) to constrain the molecular graph generation. To approximate the dipole moment of a molecule, we assume that the induced charge for a particular atom type is fixed and does not depend on the surrounding structure. We approximate these charges by learning them from the training set such that $\mathbf{Q}^*$ minimizes the $\ell_1$ loss between the actual $\mu_{dm}$ and the predicted $\hat{\mu}_{dm} = \|\mathbf{S}(i)^T\mathbf{X}(i)\mathbf{Q}^*\|_2$. Then, we consider the constraint $\hat{\mu}_{dm} \in [\xi_0, \xi_1]$.

---

[4]DruM (Jo et al., 2023) has recently shown to outperform all existing diffusion models for graph generation in their reported results. As such, PRODIGY is directly applicable to DruM, but unfortunately we are not able to report numbers on DruM as its code or pretrained models were not made publicly available by their authors at the time of submission.

Table 2: Effect of PRODIGY on constrained generic graph generation.

| | | Community-small | | | | | Ego-small | | | | | Enzymes | | | | | Grid | | | | |
|---|---|---|---|---|---|---|---|---|---|---|---|---|---|---|---|---|---|---|---|---|---|
| | | Deg↓ | Clus↓ | Orb↓ | Avg↓ | VAL$_C$↑ | Deg↓ | Clus↓ | Orb↓ | Avg↓ | VAL$_C$↑ | Deg↓ | Clus↓ | Orb↓ | Avg↓ | VAL$_C$↑ | Deg↓ | Clus↓ | Orb↓ | Avg↓ | VAL$_C$↑ |
| Edge Count | EDP-GNN | 0.362 | 0.366 | 0.125 | 0.285 | **0.23** | 0.199 | 0.469 | 0.036 | 0.235 | 0.23 | 0.117 | 0.120 | 0.004 | 0.080 | 0.56 | 1.005 | 0.033 | 0.455 | 0.498 | 0.75 |
| | +PRODIGY | 0.083 | 0.379 | 0.006 | 0.156 | 0.12 | 0.055 | 0.006 | 0.000 | 0.020 | **0.62** | 0.247 | 0.120 | 0.008 | 0.085 | **0.95** | 1.854 | 0.000 | 0.905 | 0.92 | **1.00** |
| | GDSS | 0.448 | 0.481 | 0.077 | 0.335 | 0.15 | 0.187 | 0.599 | 0.017 | 0.268 | 0.18 | 0.149 | 0.411 | 0.081 | 0.214 | 0.05 | 0.120 | 0.011 | 0.047 | 0.059 | 0.05 |
| | +PRODIGY | 0.539 | 1.096 | 0.015 | 0.550 | **0.90** | 0.104 | 0.054 | 0.001 | 0.053 | **0.65** | 0.616 | 0.966 | 0.026 | 0.536 | **0.82** | 1.249 | 0.002 | 0.604 | 0.618 | **0.95** |
| Triangle Count | EDP-GNN | 0.266 | 0.220 | 0.068 | 0.185 | 0.70 | 0.170 | 0.469 | 0.024 | 0.221 | 0.39 | 0.099 | 0.120 | 0.029 | 0.083 | 0.64 | 1.062 | 0.033 | 0.513 | 0.536 | 0.38 |
| | +PRODIGY | 0.179 | 0.595 | 0.267 | 0.347 | **0.96** | 1.340 | 0.000 | 0.018 | 0.453 | **1.00** | 1.127 | 0.000 | 0.047 | 0.391 | **1.00** | 1.996 | 0.000 | 0.978 | 0.991 | **1.00** |
| | GDSS | 0.319 | 0.187 | 0.049 | 0.185 | 0.70 | 0.160 | 0.599 | 0.005 | 0.255 | 0.32 | 0.236 | 0.222 | 0.016 | 0.158 | 0.03 | 0.154 | 0.011 | 0.050 | 0.072 | 0.00 |
| | +PRODIGY | 0.293 | 0.183 | 0.048 | 0.175 | **0.90** | 1.340 | 0.000 | 0.018 | 0.453 | **1.00** | 0.056 | 0.298 | 0.028 | 0.127 | **0.96** | 1.996 | 0.000 | 0.978 | 0.991 | **1.00** |
| Degree | EDP-GNN | 0.288 | 0.202 | 0.079 | 0.190 | 0.50 | 0.156 | 0.173 | 0.037 | 0.122 | 0.36 | 0.117 | 0.120 | 0.004 | 0.080 | 0.52 | 1.062 | 0.033 | 0.513 | 0.536 | 0.50 |
| | +PRODIGY | 0.117 | 0.726 | 0.252 | 0.365 | **0.44** | 0.042 | 0.022 | 0.000 | 0.022 | **0.63** | 0.242 | 0.000 | 0.000 | 0.081 | **1.00** | 1.717 | 0.000 | 0.958 | 0.892 | **1.00** |
| | GDSS | 0.350 | 0.203 | 0.051 | 0.201 | 0.40 | 0.131 | 0.238 | 0.018 | 0.129 | 0.32 | 0.158 | 0.217 | 0.037 | 0.137 | 0.40 | 0.154 | 0.011 | 0.050 | 0.072 | 0.00 |
| | +PRODIGY | 0.075 | 0.431 | 0.097 | 0.201 | **1.00** | 0.116 | 0.169 | 0.001 | 0.095 | **0.68** | 0.265 | 0.802 | 0.018 | 0.362 | **1.00** | 1.755 | 0.000 | 0.972 | 0.909 | **1.00** |

Table 3: Effect of PRODIGY on constrained molecular generation. OOM denotes out-of-memory.

| | | QM9 | | | | | ZINC250k | | | | |
|---|---|---|---|---|---|---|---|---|---|---|---|
| | | Val. (%) ↑ | Novel. (%) ↑ | NSPDK ↓ | FCD ↓ | VAL$_C$ ↑ | Val. (%) ↑ | Novel. (%) ↑ | NSPDK ↓ | FCD ↓ | VAL$_C$ ↑ |
| Valency | EDP-GNN | 96.95 | 76.74 | 0.01 | 6.15 | **0.97** | OOM | OOM | OOM | OOM | OOM |
| | +PRODIGY | 96.29 | 76.68 | 0.07 | 6.23 | 0.96 | OOM | OOM | OOM | OOM | OOM |
| | GDSS | 95.72 | 81.04 | 0.00 | 2.47 | 0.88 | 97.01 | 100.00 | 0.02 | 14.04 | 0.94 |
| | +PRODIGY | 99.83 | 82.74 | 0.00 | 2.82 | **0.99** | 99.88 | 100.00 | 0.09 | 29.79 | **0.99** |
| Atom Count | EDP-GNN | 96.95 | 76.74 | 0.014 | 8.63 | 0.37 | OOM | OOM | OOM | OOM | OOM |
| | +PRODIGY | 98.06 | 54.36 | 0.018 | 5.66 | **1.00** | OOM | OOM | OOM | OOM | OOM |
| | GDSS | 95.72 | 81.04 | 0.01 | 7.28 | 0.33 | 97.01 | 100.00 | 0.03 | 16.01 | 0.13 |
| | +PRODIGY | 95.02 | 67.67 | 0.00 | 1.60 | **1.00** | 96.90 | 100.00 | 0.04 | 12.24 | **0.99** |
| Molecular Weight | EDP-GNN | 96.95 | 76.74 | 0.33 | 16.86 | 0.00 | OOM | OOM | OOM | OOM | OOM |
| | +PRODIGY | 79.26 | 70.05 | 0.21 | 3.80 | **0.17** | OOM | OOM | OOM | OOM | OOM |
| | GDSS | 95.72 | 81.04 | 0.31 | 17.08 | 0.00 | 97.01 | 100.00 | 0.02 | 11.15 | 0.62 |
| | +PRODIGY | 99.95 | 81.14 | 0.18 | 4.16 | **0.53** | 97.66 | 100.00 | 0.01 | 12.15 | **0.63** |

Table 4 shows the results for unconditional and controlled settings by using PRODIGY on GeoLDM (Xu et al., 2023). For the former, we choose $\xi_0, \xi_1$ as the minimum and maximum dipole moment in the training set and find that our sampling method is able to preserve the stability measures.

Table 4: Results on 3D molecule generation with the constraint on the predicted dipole moment.

| | Unconditional | | Controlled |
|---|---|---|---|
| | Atom. Stability (%) ↑ | Mol. Stability (%) ↑ | MAE $\mu_{dm}$ ↓ |
| EDM (Hoogeboom et al., 2022) | 98.7 | 82.0 | 1.11 (0.04) |
| GeoLDM (Xu et al., 2023) | 98.9 | 89.4 | 1.10 (0.04) |
| +PRODIGY | 98.9 | 89.4 | **0.00** (1.15) |

For the controlled setting, we follow existing works and consider constraining the values to lie in the 2nd half of the training set. In particular, we choose $[\xi_0, \xi_1]$ to lie in the range of one standard deviation from the mean of the subset. Table 4 then shows the MAE of the generated molecules $= (1 - \text{VAL}_\mathcal{C})$ with the bias of the predicted model, i.e. MAE $(\mu, \hat{\mu}_{dm})$, in the parentheses. Even with a highly biased approximation of the dipole moment, we can control the generation within competitive errors to conditional methods that use more advanced prediction methods.

## 5.4 HOW PRODIGY AFFECTS THE DISTRIBUTION LEARNED BY THE ORIGINAL MODEL?

We first show that PRODIGY does not affect unconditional generation performance when the given constraint is chosen in such a manner that the whole test set satisfies it, *i.e.*, $\mathcal{G}_{ts} \subseteq \mathcal{C}$. In particular, for generic graphs, we consider the Edge Count constraint and set the number of edges to be within $[\min_{G \in \mathcal{G}_{ts}} |\mathbf{E}(G)|, \max_{G \in \mathcal{G}_{ts}} |\mathbf{E}(G)|]$ (see App. B.1). Table 5 shows that PRODIGY preserves the generation performance of the base models under this constraint, as computed by the MMD metrics. Thus, we find that our method is able to retrieve the samples close to the given data distribution when the constraint is chosen to subsume the given test set. We also find a similar trend in 2D molecule generation, where we constrain the molecular weights to lie within $[\min_{G \in \mathcal{G}_{ts}} |W(G)|, \max_{G \in \mathcal{G}_{ts}} |W(G)|]$. Table 6 shows that molecules sampled using PRODIGY have similar quality as the originally sampled molecules under this constraint.

## 5.5 HOW SENSITIVE IS OUR APPROACH TO PARAMETERS AND CONSTRAINT CONSTANTS?

Our method allows for an arbitrary constraint satisfaction (*i.e.* for any constraint parameter) and an interpretable tuning hyperparameter $\gamma_t$. Figure 3 compares the original GDSS sampling with PRODIGY for a range of budgets of edge count and a range of $\gamma_t$ values. This shows that our method is able to support a wide range of constraint parameters with appropriate tuning of the $\gamma_t$ values.

Table 5: Effect of PRODIGY on unconditional generation of generic graphs. We highlight the better of the original and PRODIGY sampling while comparing against other standard baselines.

| | Community-small | | | | Ego-small | | | | Enzymes | | | | Grid | | | |
|---|---|---|---|---|---|---|---|---|---|---|---|---|---|---|---|---|
| | Deg.↓ | Clus.↓ | Orb.↓ | Avg.↓ | Deg.↓ | Clus.↓ | Orb.↓ | Avg.↓ | Deg.↓ | Clus.↓ | Orb.↓ | Avg.↓ | Deg.↓ | Clus.↓ | Orb.↓ | Avg.↓ |
| Deep-GMG (Li et al., 2018)[4] | 0.220 | 0.950 | 0.400 | 0.523 | 0.040 | 0.100 | 0.020 | 0.053 | - | - | - | - | - | - | - | - |
| Graph-RNN (You et al., 2018)[4] | 0.080 | 0.120 | 0.040 | 0.080 | 0.090 | 0.220 | 0.003 | 0.104 | 0.017 | 0.062 | 0.046 | 0.042 | 0.064 | 0.043 | 0.021 | 0.043 |
| Graph-VAE (Simonovsky and Komodakis, 2018)[4] | 0.350 | 0.980 | 0.540 | 0.623 | 0.130 | 0.170 | 0.050 | 0.117 | 1.369 | 0.629 | 0.191 | 0.730 | 1.619 | 0.0 | 0.919 | 0.846 |
| GNF (Liu et al., 2019)[4] | 0.200 | 0.200 | 0.110 | 0.170 | 0.030 | 0.100 | 0.001 | 0.044 | - | - | - | - | - | - | - | - |
| EDP-GNN (Niu et al., 2020)[5] | 0.120 | **0.071** | 0.046 | 0.079 | 0.020 | 0.043 | 0.006 | 0.023 | **1.011** | **0.791** | 0.239 | **0.681** | 1.062 | **0.033** | **0.513** | **0.536** |
| **+PRODIGY** | **0.091** | 0.094 | **0.041** | **0.075** | **0.019** | **0.028** | **0.004** | **0.017** | 1.067 | 0.815 | **0.234** | 0.705 | **1.014** | 0.126 | 0.541 | 0.560 |
| GDSS (Jo et al., 2022)[5] | 0.170 | 0.090 | 0.079 | 0.113 | **0.023** | **0.010** | **0.013** | **0.015** | 0.034 | 0.078 | 0.003 | 0.038 | 0.154 | 0.011 | 0.050 | 0.072 |
| **+PRODIGY** | **0.132** | **0.077** | **0.044** | **0.084** | 0.028 | 0.030 | **0.013** | 0.024 | **0.033** | **0.078** | **0.003** | **0.038** | **0.154** | **0.010** | **0.050** | **0.072** |

Table 6: Effect of PRODIGY on unconditional generation of 2D molecules. We bold the best sampling strategy for diffusion models and compare against other baselines. OOM denotes out-of-memory.

| | QM9 | | | | ZINC250k | | | |
|---|---|---|---|---|---|---|---|---|
| | Val. w/o corr. (%) | Novel. (%) ↑ | NSPDK ↓ | FCD ↓ | Val. w/o corr. (%) | Novel. (%) ↑ | NSPDK ↓ | FCD ↓ |
| GraphAF (Shi et al., 2020)[4] | 67 | 88.83 | 0.020 | 5.268 | 68 | 100.00 | 0.044 | 16.289 |
| MoFlow (Zang and Wang, 2020)[4] | 91.36 | 98.10 | 0.017 | 4.467 | 63.11 | 100.00 | 0.046 | 20.931 |
| GraphEBM (Liu et al., 2021)[4] | 8.22 | 97.01 | 0.030 | 6.143 | 5.29 | 100.00 | 0.212 | 35.471 |
| EDP-GNN (Niu et al., 2020) | 96.95 | 76.74 | **0.005** | **6.151** | OOM | OOM | OOM | OOM |
| **+PRODIGY** | **97.01** | **77.12** | **0.005** | 6.187 | OOM | OOM | OOM | OOM |
| GDSS (Jo et al., 2022) | **95.72** | 86.27 | **0.003** | 2.900 | **97.01** | 100.00 | 0.019 | **14.656** |
| **+PRODIGY** | 95.22 | 83.62 | **0.003** | **2.745** | 95.61 | 100.00 | **0.014** | 15.298 |

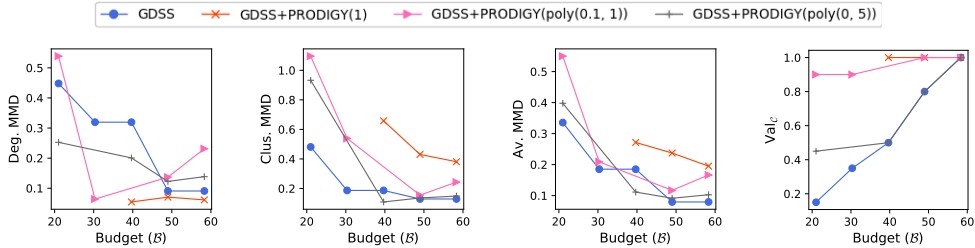

Figure 3: Comparison of methods in generating Community graphs with an arbitrary $\mathcal{B}$ number of edges. Lower MMD scores and higher constraint validity $\text{VAL}_\mathcal{C}$ are desired. We compare different values of $\gamma_t$ parameter from Section 4. Note that the lack of a data point is when sampling leads to a trivial solution of zero edges or an empty graph.

As mentioned in Section 4, we can note a trade-off that while increasing $\gamma_t$ leads to higher constraint validity ($\text{VAL}_\mathcal{C}$), it may (slightly) negatively affect the distance of the graphs from the test set (*i.e.*, an increase in the MMD scores). This means that one must choose $\gamma_t$ that is able to obtain the best of both worlds, which is provided by our method. We also find that choosing a higher power for polynomial scheduling reduces the constraint validity as the sampling favors the reverse diffusion process for the majority of diffusion timesteps except at the end. Refer Appendix D.4 for elaborate results on other datasets and constraints.

## 6 DISCUSSION AND CONCLUSION

We proposed PRODIGY, a plug-and play approach to controllable graph generation with diffusion models. Our work enables precise control of the graph generation process under arbitrary well specified and hard constraints, thereby making it applicable to wide range of real-world applications in practice including network design, drug discovery and many more. We hope that this opens future research avenues for enabling interpretable control in the generative models across different domains. Future directions include extending our methods for constrained graph generation to discrete diffusion models and enabling control of more complex non-linear properties of the graphs, *e.g.* GNN-based molecular property prediction (we show in App. B that it is non-trivial to extend our current approach to such non-linear functions).

---

[4]The values for these methods are taken directly from their papers or GDSS

[5]We could not reproduce the results for EDP-GNN and GDSS as reported in their papers.

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

APPENDIX

# A  PROJECTION OPERATORS

In this section, we discuss projection operators for the constraints mentioned in Table 1. We first solve for $\varphi_{\boldsymbol{\mu}}^A = \mathbf{Z}_A^*$ and $\varphi_{\boldsymbol{\mu}}^X = \mathbf{Z}_X^*$. Then, we propose a way to solve $\mathbf{h}_{\mathcal{C}}(\varphi_{\boldsymbol{\mu}}(\mathbf{G})) = \mathbf{0}$. Further, we replace $\boldsymbol{\mu}_0$ with $\boldsymbol{\mu}$ in the Lagrangian without loss of generality.

**KKT conditions.**  The optimal solution $\mathbf{Z}^*$ for the problem must satisfy the following conditions:

1. *Stationarity.* $\nabla_{\mathbf{Z}}\mathcal{L}|_{\mathbf{Z}^*} = \mathbf{0} \implies \mathbf{Z}_X^* - \mathbf{X} + \mu_0\nabla_{\mathbf{Z}_X}\mathbf{h}_{\mathcal{C}}(\mathbf{Z}_X^*, \mathbf{Z}_A^*) + \boldsymbol{\mu}_1 - \boldsymbol{\mu}_2 = \mathbf{0}$ such that $\mathbf{Z}_X^* \in [\mathbf{X}_m, \mathbf{X}_M]$ and $\mathbf{Z}_A^* - \mathbf{A} + \mu_0\nabla_{\mathbf{Z}_A}\mathbf{h}_{\mathcal{C}}(\mathbf{Z}_X^*, \mathbf{Z}_A^*) + \boldsymbol{\mu}_3 - \boldsymbol{\mu}_4 + \boldsymbol{\Lambda} = \mathbf{0}$ such that $\mathbf{Z}_A^* \in [\mathbf{A}_m, \mathbf{A}_M]$ and $\Lambda[i, j] = \lambda_{ij}$ if $i > j$, $\lambda_i$ if $i = j$, and $-\lambda_{ij}$ otherwise. It is hard to solve this system of equations simultaneously as $\nabla\mathbf{h}_{\mathcal{C}}$ can be non-linear so we assume either $\nabla_{\mathbf{Z}_A}\mathbf{h}_{\mathcal{C}}(\mathbf{Z}_X^*, \mathbf{Z}_A^*) = \mathbf{0}$ or $\nabla_{\mathbf{Z}_X}\mathbf{h}_{\mathcal{C}}(\mathbf{Z}_X^*, \mathbf{Z}_A^*) = \mathbf{0}$ depending on the form of $\mathbf{h}_{\mathcal{C}}(\mathbf{X}, \mathbf{A})$.

2. *Primal and Dual feasibility.* $\boldsymbol{\mu}_0, \boldsymbol{\mu}_1, \boldsymbol{\mu}_2, \boldsymbol{\mu}_3, \boldsymbol{\mu}_4 \geq \mathbf{0}$, $\mathbf{h}_{\mathcal{C}}(\mathbf{Z}_X^*, \mathbf{Z}_A^*) \leq \mathbf{0}$, $\mathbf{Z}_X^* \in [\mathbf{X}_m, \mathbf{X}_M]$, $\mathbf{Z}_A^* \in [\mathbf{A}_m, \mathbf{A}_M]$, $(\mathbf{Z}_A^*)^T = \mathbf{Z}_A^*$, $\mathrm{Diag}(\mathbf{Z}_A^*) = \mathbf{0}$.

3. *Complementary Slackness (CS).* $\boldsymbol{\mu}_0\mathbf{h}_{\mathcal{C}}(\mathbf{Z}^*) = \mathbf{0}$, $\boldsymbol{\mu}_1(\mathbf{Z}_X^* - \mathbf{X}_m) = \mathbf{0}$, $\boldsymbol{\mu}_2(\mathbf{X}_M - \mathbf{Z}_X^*) = \mathbf{0}$, $\boldsymbol{\mu}_3(\mathbf{Z}_A^* - \mathbf{A}_m) = \mathbf{0}$, $\boldsymbol{\mu}_4(\mathbf{A}_M - \mathbf{Z}_A^*) = \mathbf{0}$.

First, we note that $\boldsymbol{\mu}_0\mathbf{h}_{\mathcal{C}}(\mathbf{Z}^*) = \mathbf{0}$, $\boldsymbol{\mu}_0 \geq \mathbf{0}$, and $\mathbf{h}_{\mathcal{C}}(\mathbf{Z}^*) \leq \mathbf{0}$ imply that if $\mathbf{h}_{\mathcal{C}}(\mathbf{Z}^*(\boldsymbol{\mu}_0 = \mathbf{0})) \leq \mathbf{0}$ then $\boldsymbol{\mu}_0 = \mathbf{0}$ otherwise we find $\boldsymbol{\mu}_0 \geq \mathbf{0}$ such that $\mathbf{h}_{\mathcal{C}}(\mathbf{Z}^*(\boldsymbol{\mu}_0)) = \mathbf{0}$.

We also note that $\boldsymbol{\mu}_{1,2}$ can be replaced by a clamp operation $P_{[\cdot,\cdot]}$ that clamps $\mathbf{Z}_X^*$ within $[\mathbf{X}_m, \mathbf{X}_M]$. This is because if a certain entry of $\mathbf{Z}_X^*$ is within the range, then, the corresponding $\mu = 0$ (due to CS), and if not, we add/subtract a $\mu \geq 0$ such that $\mathbf{Z}_X^* = \mathbf{X}_m$ or $\mathbf{X}_M$ (CS). Similarly, we also note that $\boldsymbol{\mu}_{3,4}$ can be replaced by a clamp operation that clamps $\mathbf{Z}_A^*$ within $[\mathbf{A}_m, \mathbf{A}_M]$.

Thus, we can find $\mathbf{Z}_X^*$ and $\mathbf{Z}_A^*$ as $\Pi_{\mathcal{C}}(\mathbf{G}) = \varphi_{\mathbf{0}}(\mathbf{G})$ if $\mathbf{h}_{\mathcal{C}}(\varphi_{\mathbf{0}}(\mathbf{G})) \leq \mathbf{0}$, otherwise $\varphi_{\boldsymbol{\mu}}(\mathbf{G})$ such that $\mathbf{h}_{\mathcal{C}}(\varphi_{\boldsymbol{\mu}}(\mathbf{G})) = \mathbf{0}$. Here, $\varphi_{\boldsymbol{\mu}} = (\varphi_{\boldsymbol{\mu}}^X, \varphi_{\boldsymbol{\mu}}^A)$ can be found for the following two cases:

**1.** $\nabla_{\mathbf{Z}_A}\mathbf{h}_{\mathcal{C}}(\mathbf{Z}_X^*, \mathbf{Z}_A^*) = \mathbf{0}$: We get $\mathbf{Z}_A^* = P_{[\mathbf{A}_m, \mathbf{A}_M]}(\mathbf{A} - \boldsymbol{\Lambda})$ such that $(\mathbf{Z}_A^*)^T = \mathbf{Z}_A^*$, $\mathrm{Diag}(\mathbf{Z}_A^*) = \mathbf{0}$. We assume that the input $\mathbf{A}$ is undirected and has no self-loops, then, $\boldsymbol{\Lambda} = \mathbf{0}$ would be feasible. Thus, we get $\mathbf{Z}_A^* = \varphi_{\boldsymbol{\mu}}^A(\mathbf{G}) = P_{[\mathbf{A}_m, \mathbf{A}_M]}(\mathbf{A})$. We can find $\varphi_{\boldsymbol{\mu}}^X$ by solving for $\mathbf{Z}_X^*$ in the equation $\mathbf{Z}_X^* + \mu_0\nabla_{\mathbf{Z}_X}\mathbf{h}_{\mathcal{C}}(\mathbf{Z}_X^*, \mathbf{Z}_A^*) = \mathbf{X}$ and then, clamping it within $[\mathbf{X}_m, \mathbf{X}_M]$.

**2.** $\nabla_{\mathbf{Z}_X}\mathbf{h}_{\mathcal{C}}(\mathbf{Z}_X^*, \mathbf{Z}_A^*) = \mathbf{0}$: We get $\mathbf{Z}_X^* = \varphi_{\boldsymbol{\mu}}^X(\mathbf{G}) = P_{[\mathbf{X}_m, \mathbf{X}_M]}(\mathbf{X})$. We can find $\varphi_{\boldsymbol{\mu}}^A$ by solving for $\mathbf{Z}_A^*$ in the equation $\mathbf{Z}_A^* + \mu_0\nabla_{\mathbf{Z}_A}\mathbf{h}_{\mathcal{C}}(\mathbf{Z}_X^*, \mathbf{Z}_A^*) + \boldsymbol{\Lambda} = \mathbf{A}$ and then, clamping it within $[\mathbf{A}_m, \mathbf{A}_M]$, while satisfying $(\mathbf{Z}_A^*)^T = \mathbf{Z}_A^*$ and $\mathrm{Diag}(\mathbf{Z}_A^*) = \mathbf{0}$.

## A.1  EDGE COUNT ($|\mathbf{E}| \leq \mathcal{B}$)

***Find $\varphi_{\boldsymbol{\mu}}$.*** We have $\mathbf{h}_{\mathcal{C}}(\mathbf{Z}_X, \mathbf{Z}_A) = h_{\mathcal{C}}(\mathbf{Z}_A) = \frac{1}{2}\mathbf{1}^T\mathbf{Z}_A\mathbf{1} - \mathcal{B}$, $\mathbf{Z}_A \in [0, 1]$, $\mathrm{Diag}(\mathbf{Z}_A) = \mathbf{0}$, $\mathbf{Z}_A^T = \mathbf{Z}_A$. Then, we can note that $\nabla_{\mathbf{Z}_X}h_{\mathcal{C}} = \mathbf{0}$. Thus, we solve for $\mathbf{Z}_A^*$ in $\mathbf{Z}_A^* + \mu\nabla_{\mathbf{Z}_A}h_{\mathcal{C}}(\mathbf{Z}_A^*) + \boldsymbol{\Lambda} = \mathbf{A}$. Since $\nabla_{\mathbf{Z}_A}h_{\mathcal{C}} = \frac{1}{2}\mathbf{1}\mathbf{1}^T$, we get $\mathbf{Z}_A^* = \mathbf{A} - \frac{\mu}{2}\mathbf{1}\mathbf{1}^T - \boldsymbol{\Lambda}$. Satisfying $\mathrm{Diag}(\mathbf{Z}_A^*) = \mathbf{0}$, $(\mathbf{Z}_A^*)^T = \mathbf{Z}_A^*$ (given these conditions hold for $\mathbf{A}$) implies $\Lambda_{ii} = -1/2$ and $\Lambda_{ij} = \Lambda_{ji} = 0$. In other words, $\boldsymbol{\Lambda} = \mathbf{I}/2$. Thus, $\mathbf{Z}_A^* = \mathbf{A} - \mu/2\mathbf{1}\mathbf{1}^T + \mu/2\mathbf{I}$ followed by clamping between $[0, 1]$.

***Find $\mu$.*** To find $\mu$, we can do a bisection method between $\max\{0, 2(\min(\mathbf{A}) - 1)\}$ and $2\max(\mathbf{A})$. This is because $\frac{1}{2}\mathbf{1}^T P_{[0,1]}(\mathbf{A} - (\min(\mathbf{A}) - 1)\mathbf{1}\mathbf{1}^T + (\min(\mathbf{A}) - 1))\mathbf{1} = \binom{|\mathcal{V}|}{2} \geq \mathcal{B}$ and $\frac{1}{2}\mathbf{1}^T P_{[0,1]}(\mathbf{A} - \max(\mathbf{A})\mathbf{1}\mathbf{1}^T + \max(\mathbf{A})\mathbf{I})\mathbf{1} = \frac{1}{2}\mathbf{1}^T\mathbf{0}\mathbf{1} = 0 \leq \mathcal{B}$.

***Complexity.*** The bisection method finishes in $O(\log(\max(\mathbf{A}) - \max\{0, (\min(\mathbf{A}) - 1)\})/\xi) = O(\log(\frac{1}{\xi}))$ for a tolerance level $\xi$, since $\mathbf{A} \in [0, 1]$. Finding $\mathbf{Z}_A^*$ involves only matrix operations (addition) that have been highly optimized in Pytorch with the worst-case time complexity of $O(n^2)$. Thus, we get the time complexity of the projection operator as $O(n^2\log(\frac{1}{\xi}))$.

## A.2 TRIANGLE COUNT ($|\triangle| = \frac{1}{6}\mathrm{TR}(\mathbf{A}^3) \leq T$)

***Find $\varphi_\mu$.*** We have $\mathbf{h}_\mathcal{C}(\mathbf{Z}_X, \mathbf{Z}_A) = h_\mathcal{C}(\mathbf{Z}_A) = \frac{1}{6}\mathrm{tr}(\mathbf{Z}_A^3) - T$, $\mathbf{Z}_A \in [0, 1]$, $\mathrm{Diag}(\mathbf{Z}_A) = \mathbf{0}$, $\mathbf{Z}_A^T = \mathbf{Z}_A$. Then, we can note that $\nabla_{\mathbf{Z}_X} h_\mathcal{C} = \mathbf{0}$. Thus, we solve for $\mathbf{Z}_A^*$ in $\mathbf{Z}_A^* + \mu \nabla_{\mathbf{Z}_A} h_\mathcal{C}(\mathbf{Z}_A^*) + \mathbf{\Lambda} = \mathbf{A}$. Since $\nabla_{\mathbf{Z}_A} h_\mathcal{C} = \frac{1}{2}\mathbf{Z}_A^2$, we get $\mathbf{Z}_A^* + \frac{\mu}{2}(\mathbf{Z}_A^*)^2 + \mathbf{\Lambda} = \mathbf{A}$. Satisfying $\mathrm{Diag}(\mathbf{Z}_A^*) = \mathbf{0}$, $(\mathbf{Z}_A^*)^T = \mathbf{Z}_A^*$ (given these hold for $\mathbf{A}$) implies $\mathbf{\Lambda} = \mathbf{0}$. Thus, $\mathbf{Z}_A^* + \frac{\mu}{2}(\mathbf{Z}_A^*)^2 = \mathbf{A}$. Let us assume $\mathbf{Z}_A^* \approx \mathbf{A}^2$, *i.e.*, the squared values do not change a lot after projection. Then, we get $\mathbf{Z}_A^* \approx \mathbf{A} - \frac{\mu}{2}\mathbf{A}^2$.

***Find $\mu$.*** We will find for $\mu$ using the bisection method here as well. But it is non-trivial to obtain two points for which $\frac{1}{6}\mathrm{tr}(P_{[0,1]}((\mathbf{A} - \frac{\mu}{2}\mathbf{A}^2)^3)) - T$ have opposite signs. Thus, we assume that one such point is $\mu = 0$ and search for the first point $> 0$ with an opposite sign using a linear search from $\mu$ with a fixed step size $s$. Then, we apply the bisection method between $0$ and the new point $\mu_1$ found using the linear search.

***Complexity.*** Linear search computes $\mathbf{Z}_A^*$ for $(\mu_1 - 0)/s$ times to compare with value at $\mu = 0$. The bisection method finishes in $O(\log(\mu_1/\xi))$ time. Again, finding $\mathbf{Z}_A^*$ involves only matrix operations (addition) that have been highly optimized in Pytorch with the worst-case time complexity of $O(n^3)$. Thus, we get the time complexity of the projection operator as $O(n^3(\mu_1/s + \log(\mu_1/\xi)))$.

## A.3 MAX DEGREE ($d_{\mathrm{MAX}} = \mathbf{A}\mathbf{1} \leq \delta_d \mathbf{1}$)

***Find $\varphi_\mu$.*** We have $\mathbf{h}_\mathcal{C}(\mathbf{Z}_X, \mathbf{Z}_A) = \mathbf{Z}_A\mathbf{1} - \delta_d\mathbf{1}$, $\mathbf{Z}_A \in [0, 1]$, $\mathrm{Diag}(\mathbf{Z}_A) = \mathbf{0}$, $\mathbf{Z}_A^T = \mathbf{Z}_A$. Then, we can note that $\nabla_{\mathbf{Z}_X} \mathbf{h}_\mathcal{C} = \mathbf{0}$ and we solve for $\mathbf{Z}_A^*$ in $\mathbf{Z}_A^* + \boldsymbol{\mu} \cdot \nabla_{\mathbf{Z}_A} \mathbf{h}_\mathcal{C}(\mathbf{Z}_A^*) + \mathbf{\Lambda} = \mathbf{A}$. In other words, for each row $i$, we get $\mathbf{Z}_A^*[i, :] = \mathbf{A}[i, :] - \mu_i\mathbf{1} - \mathbf{\Lambda}[i, :]$ since $\nabla_{\mathbf{Z}_A} h_\mathcal{C}^{(i)} = \mathbf{1}$. Due to symmetricity, we obtain $\mathbf{A}[i, j] - \mu_i - \mathbf{\Lambda}[i, j] = \mathbf{A}[j, i] - \mu_j - \mathbf{\Lambda}[j, i]$ for all $i, j$, which gives us $\mu_i + \mathbf{\Lambda}[i, j] = \mu_j + \mathbf{\Lambda}[j, i]$. We can thus let $\mathbf{\Lambda}[i, j] = \frac{1}{2}(\mu_j - \mu_i)$ for all $i \neq j$. For the diagonal entries, we want $\mathbf{\Lambda}[i, i] = -\mu_i$ so that $\mathbf{Z}_A^*$ has no non-zero diagonal entries. Thus, we get $\mathbf{Z}_A^* = \mathbf{A} - \frac{1}{2}(\boldsymbol{\mu}\mathbf{1}^T + \mathbf{1}\boldsymbol{\mu}^T) + \mathrm{Diag}(\boldsymbol{\mu})$ followed by clamping between $[0, 1]$.

***Find $\mu$.*** Since $\mathbf{h}_\mathcal{C}$ is a vector function, we cannot find its root using the bisection method. Instead, we divide $\mathbf{h}_\mathcal{C}(\varphi_{\boldsymbol{\mu}}) = \mathbf{0}$ into multiple equations $\tilde{h}_\mathcal{C}^{(i)}(\varphi_{\mu_i}) = 0$ that we can solve independently such that the root $\tilde{\boldsymbol{\mu}}$ obtained by concatenating these $\tilde{\mu}_i$s satisfies $\mathbf{h}_\mathcal{C}(\varphi_{\tilde{\boldsymbol{\mu}}}) \leq \mathbf{0}$.

In particular, we can just solve each row $\mu_i$'s equation separately and add it later to satisfy the symmetricity. Thus, we have to solve for $\tilde{\mu}_i \geq 0$ such that $\mathbf{1}^T P_{[0,1]}(\mathbf{A}[i, :] - \tilde{\mu}_i\mathbf{1}) = \delta_d$. Thus, we solve for $\tilde{\mu}_i$ for all $i$ and use it to find $\tilde{\boldsymbol{\mu}}$ using the bisection method between $\max\{0, 2(\min(\mathbf{A}[i, :]) - 1)\}$ and $2\max(\mathbf{A}[i, :])$ (due to the same logic as for Edge Count constraint). Note that if $\mathbf{1}^T P_{[0,1]}(\mathbf{A}[i, :] - \tilde{\mu}_i\mathbf{1}) = \delta_d$, then $\mathbf{1}^T P_{[0,1]}(\mathbf{A}[i, :] - (\tilde{\mu}_i + \epsilon)\mathbf{1}) \leq \delta_d$, for all $\epsilon \geq 0$, because $(\tilde{\mu}_i + \epsilon) \geq \tilde{\mu}_i$ and it is a decreasing function. We have $\epsilon_i$ to be $\tilde{\mu}_j$ for different columns $j$. Thus, $P_{[0,1]}(\mathbf{A} - \frac{2}{2}(\tilde{\boldsymbol{\mu}}\mathbf{1}^T + \mathbf{1}\tilde{\boldsymbol{\mu}}^T) + 2\mathrm{Diag}(\tilde{\boldsymbol{\mu}}))\mathbf{1} \leq \delta_d\mathbf{1}$.

***Complexity.*** We solve $n$ different equations using the bisection method in time $O(\log(\frac{1}{\xi}))$ as $\mathbf{A} \in [0, 1]$. Note that this can be done in a parallel manner by using the Pytorch functionalities. Again, finding $\mathbf{Z}_A^*$ involves only matrix addition that has been highly optimized in Pytorch with the worst-case time complexity of $O(n^2)$. Thus, we get the time complexity of the projection operator as $O(n^2 \cdot n\log(1/\xi)) = O(n^3\log(1/\xi))$.

## A.4 VALENCY ($\mathbf{A}\mathbf{1} \leq \mathbf{X}\mathbf{v}$)

Here, we fix $\mathbf{X}$ and let $\mathbf{X}\mathbf{v} = \mathbf{u}$ denote the weighted valency of each node in the graph. Then, the constraint becomes similar to the Max Degree constraint and we follow the same steps to find $\mathbf{Z}_A^* = \mathbf{A} - \frac{1}{2}(\boldsymbol{\mu}\mathbf{1}^T + \mathbf{1}\boldsymbol{\mu}^T) + \mathrm{Diag}(\boldsymbol{\mu})$ except now, we clamp within $[0, 3]$ since it's a molecular graph and clamp $\mathbf{X}$ within $[0, 1]$ as well.

## A.5 ATOM COUNT ($\mathbf{X}^T\mathbf{1} - \mathbf{c}$)

***Find $\varphi_\mu$.*** We have $\mathbf{h}_\mathcal{C}(\mathbf{Z}_X, \mathbf{Z}_A) = \mathbf{Z}_X^T\mathbf{1} \leq \mathbf{c}$, $\mathbf{Z}_X \in [0, 1]$. Then, we can note that $\nabla_{\mathbf{Z}_A}\mathbf{h}_\mathcal{C}(\mathbf{Z}_X, \mathbf{Z}_A) = \mathbf{0}$ and for each column or atom type in $\mathbf{X}$, we get $\mathbf{Z}_X^*[:, j] = \mathbf{X}[:, j] - \mu_j\mathbf{1}^T$ since $\nabla_{\mathbf{Z}_X}\mathbf{h}_\mathcal{C} = \mathbf{1}$. Thus, we get $\mathbf{Z}_X^* = P_{[0,1]}(\mathbf{X} - \mathbf{1}\boldsymbol{\mu}^T)$.

***Find $\mu$.*** Since $\mathbf{h}$ is a vector-valued function, we cannot obtain its root w.r.t. $\mu$ directly using the bisection method. However, we make another observation that allows us to do that. In particular, $\mathbf{h}_{\mathcal{C}}(\varphi_{\mu}) = \mathbf{0}$ can be divided into $F$ independent equations such that $h_{\mathcal{C}}^j$ satisfies the $j$th column $(\mathbf{Z}_X^*[:,j] - \mu_j \mathbf{1}^T)\mathbf{1} = c_j$. This can be solved independently for each $j$ using the bisection method between $[\max\{0, \min_i(X_{ij}) - 1\}, \max_i(X_{ij})]$ as $\sum_i P_{[0,1]}(X_{ij} - \max_i(X_{ij})) = 0 \leq c_j$ and $\sum_i P_{[0,1]}(X_{ij} - \min_i(X_{ij}) + 1) = |\mathcal{V}| \geq c_j$.

***Complexity.*** We solve $F$ different equations using bisection method with $\log(\frac{1}{\xi})$ steps each, as $\mathbf{X} \in [\mathbf{0}, \mathbf{1}]$. Further, $\varphi_{\mu}^X$ only involves a matrix addition that is almost constant in Pytorch with worst-case complexity of $O(n^2)$. The total complexity thus, becomes $O(n^2 F \log(\frac{1}{\xi}))$.

## A.6 MOLECULAR WEIGHT ($\mathbf{1}^T \mathbf{Xm} \leq W$)

***Find $\varphi_{\mu}$.*** We have $\mathbf{h}_{\mathcal{C}}(\mathbf{Z}_X, \mathbf{Z}_A) = h_{\mathcal{C}}(\mathbf{Z}_X, \mathbf{Z}_A) = \mathbf{1}^T \mathbf{Z}_X \mathbf{m} \leq W, \mathbf{Z}_X \in [\mathbf{0}, \mathbf{1}]$. Then, $\nabla_{\mathbf{Z}_A} h_{\mathcal{C}} = \mathbf{0}$ and $\nabla_{\mathbf{Z}_X} h_{\mathcal{C}}(\mathbf{Z}_X, \mathbf{Z}_A) = \mathbf{1m}^T$, which gives us $\mathbf{Z}_X^* = \mathbf{X} - \mathbf{1}\mu^T$ followed by clamping within $[\mathbf{0}, \mathbf{1}]$.

***Find $\mu$.*** It is non-trivial to find two end-points between which we can conduct the bisection method for $\mathbf{1}^T P_{[0,1]}(\mathbf{X} - \mathbf{1}\mu^T)\mathbf{m} = W$. Thus, we assume that one such point is $\mu = 0$ and search for the first point $> 0$ with an opposite sign using a linear search from $\mu$ with a fixed step size $s$. Then, we apply the bisection method between $0$ and the new point $\mu_1$ found using the linear search.

***Complexity.*** Linear search finds $\varphi_{\mu}^X$ for $\mu_1/s$ different values of $\mu$. This is followed by a bisection method that finishes in $O(\log(\mu_1/\xi))$ steps. Computing $\varphi_{\mu}^X$ involves just matrix addition that has been highly optimized in Pytorch with worst-case complexity of $O(n^2)$. Thus, the total time-complexity of the projection operator can be given as $O(n^2(\mu_1/s + \log(\mu_1/\xi)))$.

## A.7 DIPOLE MOMENT ($\|\mathbf{S}^T \mathbf{XQ}\|_2 \in [\xi_0, \xi_1]$)

A 3D molecular structure can be described as $(\mathbf{X}, \mathbf{S})$, where $\mathbf{S} \in \mathbb{R}^{n \times 3}$ denotes the positions of each atom in the 3-dimensional space from the center-of-mass origin. In addition to this structure, one also requires other quantitative measures such as atomic charges to calculate molecular properties. Let the charges in a molecule be given by a vector $\mathbf{Q} \in \mathbb{R}^F$, then the dipole moment vector can be written as $\mu_{dm} = \mathbf{S}^T \mathbf{XQ} \in \mathbb{R}^3$. We consider a constraint on its norm as $\|\mu_{dm}\|_2 = \|\mathbf{S}^T \mathbf{XQ}\|_2 \in [\xi_0, \xi_1]$. We assume no projection with respect to $\mathbf{X}$ and project $\mathbf{S}$ using the projection vector of the $\ell_2$-norm (Parikh et al., 2014) simply as $\mathbf{Z}_S^* = \mathbf{S}$ if $\|\mathbf{S}^T \mathbf{XQ}\|_2 \in [\xi_0, \xi_1]$ otherwise $\mathbf{S}/\|\mathbf{S}^T \mathbf{XQ}\|_2 \cdot \xi_0$ if $\|\mathbf{S}^T \mathbf{XQ}\|_2 < \xi_0$ and $\mathbf{S}/\|\mathbf{S}^T \mathbf{XQ}\|_2 \cdot \xi_1$ if $\|\mathbf{S}^T \mathbf{XQ}\|_2 > \xi_1$.

However, note that the charges on each atom are unknown and depend upon the given molecular structure. As an approximation, we learn the charges for each atom $\mathbf{Q}$ from the dataset by minimizing the $\ell_1$ loss $\sum_{i \in \mathcal{D}} \left| \mu_{dm}(i) - \|\mathbf{S}(i)^T \mathbf{X}(i)\mathbf{Q}\|_2 \right|$ over the trainable parameters $\mathbf{Q} \in \mathbb{R}^F$.

# B EXTENSIONS

In this section, we discuss several extensions and the corresponding recipes to support more complex constraints and properties including, box constraints and linear and non-linear properties.

## B.1 BOX CONSTRAINT

We note that our formulation allows us to solve a box constraint from the projection operator for the upper bound constraint. In particular, a box constraint can be defined as $\mathcal{C} = \{\mathbf{G} : \delta_{low} \leq b(\mathbf{G}) \leq \delta_{upp}\}$. This is equivalent to considering $\mathbf{h}_{\mathcal{C}} : [h_{\mathcal{C}}^1, h_{\mathcal{C}}^2]$, such that $h_{\mathcal{C}}^1(\mathbf{G}) = \delta_{low} - b(\mathbf{G})$ and $h_{\mathcal{C}}^2(\mathbf{G}) = b(\mathbf{G}) - \delta_{upp}$. Given that $\delta_{low} \leq \delta_{upp}$, we can note that both $h_{\mathcal{C}}^1(\mathbf{G}) > 0$ and $h_{\mathcal{C}}^2(\mathbf{G}) > 0$ cannot hold. Thus, we get

$$\Pi_{\mathcal{C}}(G) = \begin{cases} \varphi_0(\mathbf{G}) & ; h_{\mathcal{C}}^1(\varphi_0(\mathbf{G})) \leq 0, h_{\mathcal{C}}^2(\varphi_0(\mathbf{G})) \leq 0 \\ \varphi_{\mu}(\mathbf{G}) & ; h_{\mathcal{C}}^1(\varphi_0(\mathbf{G})) \leq 0, h_{\mathcal{C}}^2(\varphi_0(\mathbf{G})) > 0, h_{\mathcal{C}}^2(\varphi_{\mu}(\mathbf{G})) = 0 \\ \varphi_{-\mu}(\mathbf{G}) & ; h_{\mathcal{C}}^2(\varphi_0(\mathbf{G})) \leq 0, h_{\mathcal{C}}^1(\varphi_0(\mathbf{G})) > 0, h_{\mathcal{C}}^1(\varphi_{-\mu}(\mathbf{G})) = 0 \end{cases} \tag{5}$$

## B.2 Linear Property Approximators ($\mathbf{1}^T \hat{\mathbf{A}}^k \mathbf{X} \Theta \leq p$)

***Find $\varphi_\mu$.*** We have $\mathbf{h}_\mathcal{C}(\mathbf{Z}_X, \mathbf{Z}_A) = h_\mathcal{C}(\mathbf{Z}_X, \mathbf{Z}_A) = \mathbf{1}^T \hat{\mathbf{Z}}_A^k \mathbf{Z}_X \Theta - p$, $\mathbf{Z}_X \in [\mathbf{0}, \mathbf{1}]$. We fix $\mathbf{A}$ and thus, assume $\mathbf{Z}_A = P_{[\mathbf{0},\mathbf{3}]}(\mathbf{A})$. Let $\hat{P}_{[\mathbf{0},\mathbf{3}]}(\mathbf{A})$ denote the normalized adjacency matrix corresponding to $P_{[\mathbf{0},\mathbf{3}]}(\mathbf{A})$. Then, $\nabla_{\mathbf{Z}_A} h_\mathcal{C} = \mathbf{0}$ and $\nabla_{\mathbf{Z}_X} h_\mathcal{C}(\mathbf{Z}_X, \mathbf{Z}_A) = (\hat{P}_{[\mathbf{0},\mathbf{3}]}(\mathbf{A})^k)^T \mathbf{1} \Theta^T$, which gives us $\mathbf{Z}_X^* = \mathbf{X} - \mu(\hat{P}_{[\mathbf{0},\mathbf{3}]}(\mathbf{A})^k)^T \mathbf{1} \Theta^T$ followed by clamping within $[\mathbf{0}, \mathbf{1}]$.

***Find $\mu$.*** It is non-trivial to find two end-points between which we can conduct the bisection method for which there is equality on $h_\mathcal{C}$. Thus, we assume that one such point is $\mu = 0$ and search for the first point $> 0$ with an opposite sign using a linear search from $\mu$ with a fixed step size $s$. Then, we apply the bisection method between 0 and the new point $\mu_1$ found using the linear search.

***Complexity.*** Linear search finds $\varphi_\mu^X$ for $\mu_1/s$ different values of $\mu$. This is followed by a bisection method that finishes in $O(\log(\mu_1/\xi))$ steps. Computing $\varphi_\mu^X$ involves just matrix multiplication that has been highly optimized in Pytorch with worst-case complexity of $O(n^3)$. Thus, the total time-complexity of the projection operator can be given as $O(n^3(\mu_1/s + \log(\mu_1/\xi)))$.

## B.3 Non-Linear Property Approximators

Many graph properties are estimated using neural networks with non-linear activation functions. Constraining these properties implies constraining the output of these networks. Let us consider a typical single-layer neural network prediction, whose output can be written as $\mathbf{w}_2^T \text{ReLU}(\mathbf{W}_1^T \mathbf{u} + \mathbf{b}_1) + b_2$. The corresponding constraint would then look like $h_\mathcal{C}(\mathbf{u}) = \mathbf{w}_2^T \text{ReLU}(\mathbf{W}_1^T \mathbf{u} + \mathbf{b}_1) + b_2 - \epsilon \leq 0$ for some trained parameters $\mathbf{W}_1, \mathbf{w}_2, \mathbf{b}_1, b_2$. We want to find $z_u$ such that $\|\mathbf{z}_u - \mathbf{u}\|_2^2$ is minimized such that this constraint is satisfied. The Lagrangian is given as $\mathcal{L}(\mathbf{z}_u, \lambda) = \frac{1}{2}\|\mathbf{z}_u - \mathbf{u}\|_2^2 + \lambda(\mathbf{w}_2^T \text{ReLU}(\mathbf{W}_1^T \mathbf{u} + \mathbf{b}_1) + b_2 - \epsilon)$ and applying the KKT conditions give us $(\mathbf{z}_u^* - \mathbf{u}) + \lambda \mathbf{w}_2^T \mathbb{1}\{\mathbf{W}_1^T \mathbf{z}_u^* + \mathbf{b}_1 \geq 0\} \odot \mathbf{W}_1^T \mathbf{1} = \mathbf{0}, \lambda \geq 0, \mathbf{w}_2^T \text{ReLU}(\mathbf{W}_1^T \mathbf{z}_u^* + \mathbf{b}_1) + b_2 - \epsilon \leq 0, \lambda(\mathbf{w}_2^T \text{ReLU}(\mathbf{W}_1^T \mathbf{z}_u^* + \mathbf{b}_1) + b_2 - \epsilon) = 0$. If $\mathbf{w}_2^T \text{ReLU}(\mathbf{W}_1^T \mathbf{u} + \mathbf{b}_1) + b_2 - \epsilon \leq 0$, then $\mathbf{z}^* = \mathbf{u}$ (since $\lambda = 0$ otherwise we find a $\lambda \geq 0$ and $\mathbf{z}_u^*$ such that $\mathbf{w}_2^T \text{ReLU}(\mathbf{W}_1^T \mathbf{z}_u^* + \mathbf{b}_1) + b_2 - \epsilon = 0$ and $(\mathbf{z}_u^* - \mathbf{u}) + \lambda \mathbf{w}_2^T \mathbb{1}\{\mathbf{W}_1^T \mathbf{z}_u^* + \mathbf{b}_1 \geq 0\} \odot \mathbf{W}_1^T \mathbf{1} = \mathbf{0}$. Solving such a system of equations is hard since the first equation gives us $\mathbf{w}_2^T \text{ReLU}(\mathbf{W}_1^T \mathbf{z}_u^* + \mathbf{b}_1) = \epsilon - b_2$, which can have infinitely many solutions for $\text{ReLU}(\mathbf{W}_1^T \mathbf{z}_u^* + \mathbf{b}_1)$ and consequently $\mathbb{1}\{\mathbf{W}_1^T \mathbf{z}_u^* + \mathbf{b}_1 \geq 0\}$ and $\mathbf{z}_u^*$, which can not be directly substituted in the second equation. Therefore, we do not consider non-linear approximators of graph properties and leave it for future works to find efficient projections for these general functions.

## C Additional Experiment Details

### C.1 Datasets

We consider the following 4 generic graph datasets:

1. **Ego-small** contains 200 small ego graphs from larger Citeseer network Sen et al. (2008).

2. **Community-small** consists of 100 randomly generated community graphs.

3. **Enzymes** has 587 protein graphs of the enzymes from the BRENDA database Schomburg et al. (2004).

4. **Grid** is a dataset of 100 standard 2D grids.

We also consider these 2 molecular graph datasets:

1. **QM9** consists of 133k small molecules with $1 - 9$ atoms as Carbon (C), Nitrogen (N), Oxygen (O), and Fluorine (F).

2. **ZINC250k** consists of 250k molecules with $6 - 38$ atoms as Carbon (C), Nitrogen (N), Oxygen (O), Fluorine (F), Phosphorus (P), Chlorine (Cl), Bromine (Br), and Iodine (I).

Table 7: PRODIGY parameters for each setting for the GDSS in the generic graph datasets

| Constraint | Community-small | Ego-small | Enzymes | Grid |
|---|---|---|---|---|
| Edge Count | poly(0.1, 1) | poly(0.1, 5) | poly(0, 1) | poly(0, 5) |
| Triangle Count | poly(0, 1) | poly(0.1, 5) | poly(0.1, 5) | poly(0.1, 5) |
| Degree | poly(0, 1) | poly(0.1, 5) | poly(0, 1) | poly(0.1, 5) |

## C.2 HYPERPARAMETERS AND SAMPLING SETUP

For constrained generation (Section 5.2) with EDP-GNN, we just use a fixed $\gamma_t = 1$ as it only consists of a Langevin corrector and an identity predictor in the sampling stage Niu et al. (2020); Jo et al. (2022); Song et al. (2020). Since there is no predictor step, each diffusion timestep is independent and is not iterative. Thus, projecting at each step has no effect on the dynamics. For GDSS, the parameters that we used for the minimal constraint generation are provided in Table 7. On the other hand, for unconditional generation (Section 5.4), we needed a much slowly growing $\gamma_t$ for GDSS and a low fixed value for EDP-GNN. In particular, we chose $\gamma_t = \text{poly}(0, 100)$ for GDSS. Note that this becomes equivalent to doing projection only at the end, not in the intermediate timesteps. For EDP-GNN, we chose a low $\gamma_t = 0.01$.

**Standard Deviations.** We run our sampling for 3 different random seeds and find standard deviations of at most $0.05$ for all the MMD metrics and up to $0.00$ for the constraint validity metric (for a given set of parameters for the PRODIGY sampling).

**Discretization.** To obtain a discrete graph from the continuous graph at the end, we use a simple rounding scheme following existing works (Jo et al., 2022; 2023; Hoogeboom et al., 2021). In particular, we do $\lfloor \mathbf{A}_0 \rceil$ for each element to obtain the discrete adjacency matrix and $\arg\max_i \mathbf{X}_0[i]$ for attribute of each node $i$.

**Additional Evaluation Metric Detail.** We only use the test graphs that satisfy the given constraint for such computation. For generic graphs, we used degree (Deg.), clustering coefficient (Clus.), and the number of occurrences of orbits with 4 nodes (Orb.)

## D ADDITIONAL RESULTS

### D.1 HOW DOES PRODIGY COMPARE TO STATE-OF-ART CONDITIONAL (SOFT-CONTROL) GENERATION APPROACHES?

To answer this question, we compare our results with that of DiGress Vignac et al. (2022), that considers a soft constraint on the molecular properties which our approach also supports (but with the ability to apply hard interpretable constraint).

**Molecular Property.** Here, we use PRODIGY to generate molecules with a specified molecular property. We follow the framework of DiGress Vignac et al. (2022) and constrain the dipole moment ($\mu$) and the highest occupied molecular orbit (HOMO) of the generated molecules to be close to a certain set of values. These properties cannot be written easily in terms of the molecular graph ($\mathbf{X}$, $\mathbf{A}$), as required by our framework.

Table 8: MAE in Molecular Property constrained generation.

| | $\mu$ | HOMO |
|---|---|---|
| DiGress (Unconditional) | $1.71 \pm .04$ | $0.93 \pm .01$ |
| DiGress+Guidance | $0.81 \pm .04$ | $0.56 \pm .01$ |
| GDSS | $2.09 \pm .01$ | $0.30 \pm .02$ |
| **GDSS+PRODIGY** | $1.09 \pm .02$ | $0.29 \pm .10$ |

Hence, we train a simple graph convolutional network Wu et al. (2019), as described in Section 3.1, to act as a proxy for the molecular property. We then constrain the predicted property to lie within a range of the given value. Following the conditional generation of Digress, we consider minimizing the mean absolute error (MAE) between the generated molecules and the first hundred molecules of QM9. We thus use the median of these values to constrain the predicted property. Table 8 shows the performance of our sampling technique (on top of a pre-trained GDSS) model against the DiGress

baseline. We can see that even after using a simple linear model for property estimation, we can generate molecules with competitive $\mu$ and HOMO as DiGress that employs a Graph Transformer as the proxy.

**3D molecule generation.** Atomic and molecular stabilities of the generated molecules in the controlled setting are $69.11\%$ and $61.16\%$ respectively.

## D.2 RUNNING TIME

**Efficiency.** For each constraint, $\varphi_{\mu}$ involves matrix multiplication operations that can be efficiently done by exploiting data batching and parallelization of Pytorch (Paszke et al., 2019). Finding $\mu$ is also efficient as the bisection method converges in logarithmic number of steps and can exploit data batching strategies, thereby making the entire approach highly efficient.

We report the PRODIGY sampling time for different constraints on different datasets with GDSS. In particular, Table 9 reports the sampling time taken per diffusion timestep. This shows that the projection step doesn't change the scale of the diffusion sampling and the time taken is mostly minimal as compared to the denoising step. This is in line with the almost logarithmic scaling of these projection operations with respect to the number of edges.

Table 9: Time taken (in seconds) per diffusion timestep. $^{*}$ denotes the time taken by the original (unconstrained) GDSS sampling.

|  | Original* | Edge Count | Triangle Count | Degree |
|---|---|---|---|---|
| Community-small | 0.47 | 0.58 | 0.51 | 0.57 |
| Ego-small | 0.04 | 0.13 | 0.07 | 0.13 |
| Enzymes | 0.07 | 0.41 | 0.11 | 0.22 |
| Grid | 0.24 | 0.52 | 0.24 | 0.43 |

## D.3 VISUALIZATIONS

Figures 4, 5, 6, 7 compare generations of GDSS with GDSS+PRODIGY sampling on different datasets given a maximal constraint. On the other hand, generations by our method for minimal constraint for each dataset are provided in Figures 8, 9, 10, 11. We can observe that the generated graphs can satisfy the given constraints while being close to the original distribution.

Furthermore, figure 12 shows some molecules that satisfy the atom count constraint. We can note that these are quite novel structures that are produced due to the constraint of having just C and O. Rings of larger sizes are also obtained which may not be desirable. Due to the flexibility of our method to support arbitrary constraints, we believe that ring size can also be controlled. However, it is an NP-hard problem to find whether a ring would exist or not in a discrete graph structure. Continuous relaxations of this problem can be studied and then controlled for using our method but leave it as future work due to the complexity of the ring-finding problem of size larger than 3.

Figure 13 shows some sample molecules that were generated with predicted dipole moment within the specified range. This shows that we can generate molecules with a large variety in atom types as we see Oxygen and Nitrogen across different ranges.

## D.4 SENSITIVITY OF $\gamma_t$

Figures 14, 15, 16, 17 show an exhaustive analysis for different values of $\gamma_t$ and different values of the constraint parameters for different constraints. We find similar trends as Figure 3 and find a trade-off for constraint satisfaction and distance on the test set for different $\gamma_t$. We thus choose a higher power of the exponent to schedule $\gamma_t$ for unconditional generation.

### D.5 CAN WE OBTAIN LOWER MMDS FOR MINIMAL CONSTRAINED SETTING?

We find that MMDs of PRODIGY sampling in Table 2 were not very low. We note some of the issues that may be leading to this result. **(1)** Our constraint $\mathcal{P}(\mathbf{G}) \leq \min_{G \in \mathcal{G}_{ts}}\{\mathcal{P}(G)\} := \mathcal{P}_{\min}$ does not completely match the minimal test set as one can generate a graph such that $\mathcal{P}(\mathbf{G}) < \mathcal{P}_{\min}$. Hence, we now consider the constraint $\mathcal{P}(\mathbf{G}) = \mathcal{P}_{\min}$ or $\mathcal{P}_{\min} \leq \mathcal{P}(\mathbf{G}) \leq \mathcal{P}_{\min}$ when the test graphs are more than 10 (following Thompson et al. (2022)) otherwise we consider $\mathcal{P}_{\min} \leq \mathcal{P}(\mathbf{G}) \leq \mathcal{P}_{\min} + (\mathcal{P}_{\max} - \mathcal{P}_{\min})/5$. **(2)** As we see above in Appendix D.4, the parameter $\gamma_t$ controls the tradeoff between constraint satisfaction and distance with respect to certain functions from the held-out set. In Table 2, we chose this parameter with a low power of at most 5, while we now consider an extremely slow-growing $\gamma_t = \text{poly}(0, 50)$.

Table 10 shows these results for different datasets and constraints. We omit the maximum degree constraint here since it is not clear how to come up with a lower bound since it only constrains the maximum degree (not the minimum). We see a significant reduction in MMD with the constraint-filtered set and an increase in constraint validity as compared to the original GDSS. This highlights the flexibility of our method to satisfy arbitrary constraints with a flexible parameter. A curious case is the Grid graphs where we don't do much good, especially on degree and orbit MMDs for triangle count constraint. It is to be noted however that GDSS generates grids with at least one triangle, which makes these graphs invalid. On the other hand, while satisfying the triangle count equals zero constraint, we bring the clustering coefficient MMD exactly zero but this comes at the cost of hurting the degree distribution. Since GDSS is not able to generate a single valid grid, we believe this is because GDSS has learned an invalid distribution of grids.

Table 10: Effect of PRODIGY on constrained generic graph generation with both lower and upper bounds.

| | | Community-small | | | | | Ego-small | | | | | Enzymes | | | | | Grid | | | | |
|---|---|---|---|---|---|---|---|---|---|---|---|---|---|---|---|---|---|---|---|---|---|
| | | Deg.↓ | Clus.↓ | Orb.↓ | Avg.↓ | VAL$_C$↑ | Deg.↓ | Clus.↓ | Orb.↓ | Avg.↓ | VAL$_C$↑ | Deg.↓ | Clus.↓ | Orb.↓ | Avg.↓ | VAL$_C$↑ | Deg.↓ | Clus.↓ | Orb.↓ | Avg.↓ | VAL$_C$↑ |
| Edge Count | GDSS | 0.448 | **0.481** | 0.077 | 0.335 | 0.10 | 0.187 | 0.599 | 0.017 | 0.268 | 0.15 | 1.888 | 1.372 | 0.267 | 1.175 | 0.00 | **0.139** | 0.011 | **0.048** | **0.066** | **0.45** |
| | **+PRODIGY**($\gamma_t = \text{poly}(0,5)$) | 0.159 | 0.752 | 0.007 | 0.306 | **0.20** | 0.087 | 0.125 | **0.001** | 0.071 | 0.18 | **0.047** | **0.022** | 0.002 | **0.024** | **0.07** | 1.249 | **0.002** | 0.604 | 0.618 | 0.00 |
| | **+PRODIGY**($\gamma_t = \text{poly}(0,50)$) | **0.036** | 0.506 | **0.002** | **0.181** | **0.20** | **0.046** | **0.122** | **0.001** | **0.056** | **0.30** | 0.056 | 0.069 | **0.001** | 0.042 | 0.04 | 0.150 | **0.010** | 0.052 | 0.071 | **0.45** |
| Triangle Count | GDSS | **0.743** | 0.281 | **0.293** | **0.439** | 0.35 | **0.160** | 0.599 | **0.005** | **0.255** | 0.33 | 1.268 | 1.372 | 0.123 | 0.921 | 0.00 | **0.154** | 0.011 | **0.050** | **0.072** | 0.00 |
| | **+PRODIGY**($\gamma_t = \text{poly}(0,5)$) | 0.839 | **0.098** | 0.379 | 0.438 | 0.30 | 1.340 | **0.000** | 0.018 | 0.453 | **1.00** | **1.127** | **0.000** | 0.047 | **0.391** | **1.00** | 1.996 | **0.000** | 0.978 | 0.991 | **1.00** |
| | **+PRODIGY**($\gamma_t = \text{poly}(0,50)$) | 0.810 | 0.144 | 0.373 | 0.442 | **0.50** | 1.229 | **0.000** | 0.017 | 0.415 | **1.00** | **1.127** | **0.000** | **0.047** | **0.391** | **1.00** | 1.996 | **0.000** | 0.978 | 0.991 | **1.00** |

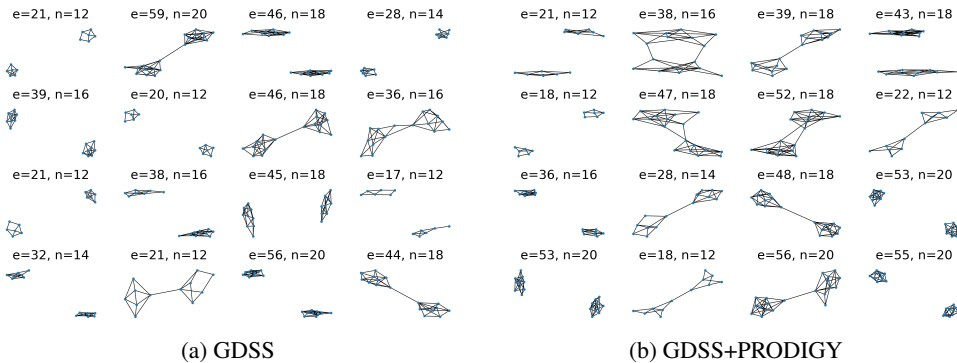

(a) GDSS                                    (b) GDSS+PRODIGY

Figure 4: Comparison of unconditional generation on Community-small dataset

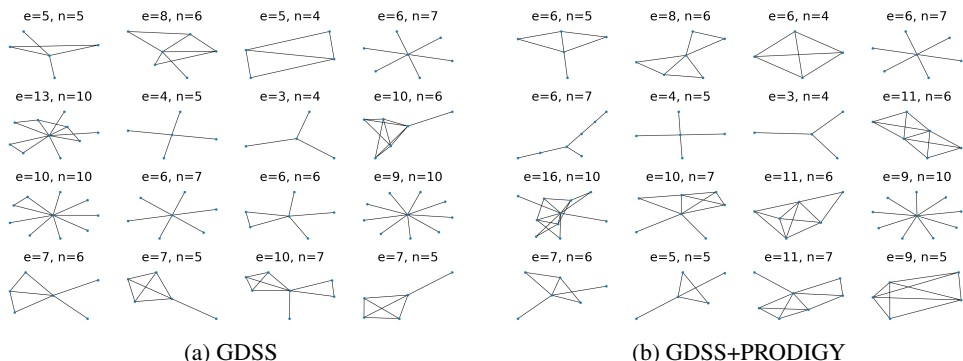

(a) GDSS                                    (b) GDSS+PRODIGY

Figure 5: Comparison of unconditional generation on Ego-small dataset

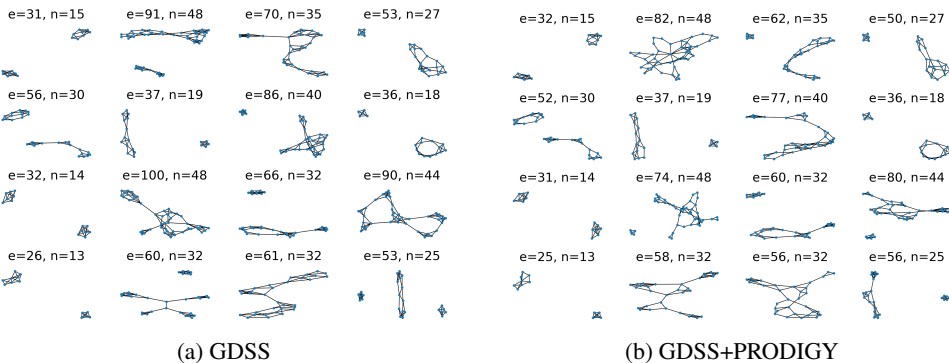

(a) GDSS                                    (b) GDSS+PRODIGY

Figure 6: Comparison of unconditional generation on Enzymes dataset

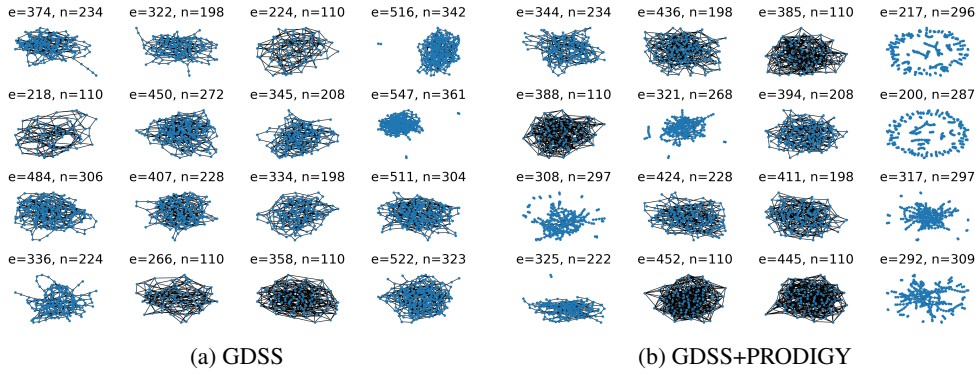

Figure 7: Comparison of maximal constraint generation on Grid dataset

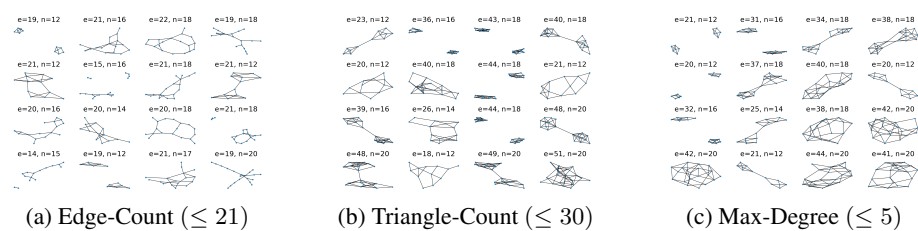

Figure 8: GDSS+PRODIGY generations for the *minimal* constrained setting on Community-small

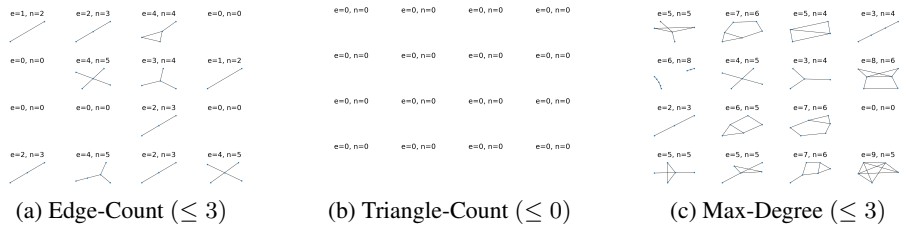

Figure 9: GDSS+PRODIGY generations for the *minimal* constrained setting on Ego-small.

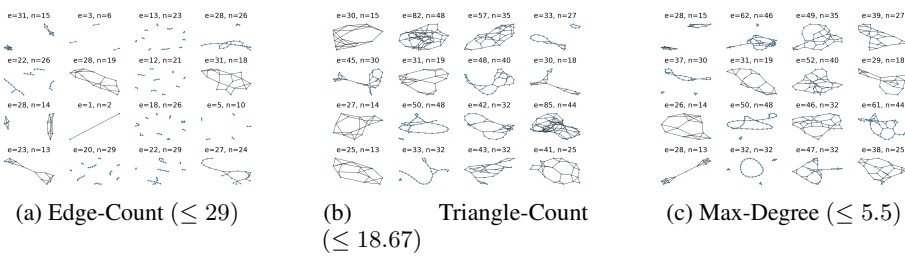

Figure 10: GDSS+PRODIGY generations for the *minimal* constrained setting on Enzymes

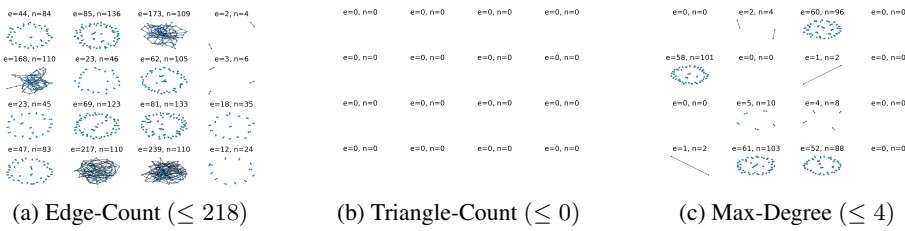

Figure 11: GDSS+PRODIGY generations for the *minimal* constrained setting on Grid

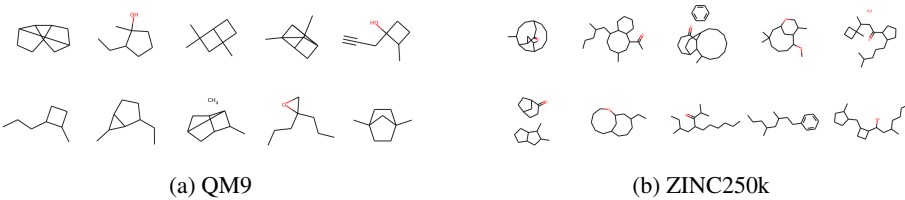

Figure 12: GDSS+PRODIGY generations for the Atom-Count constraint to generate molecules with only Carbon and Oxygen atoms. We pick the 10 novel molecules (*i.e.*, not in the dataset) with the maximum Tanimoto similarity with the test dataset.

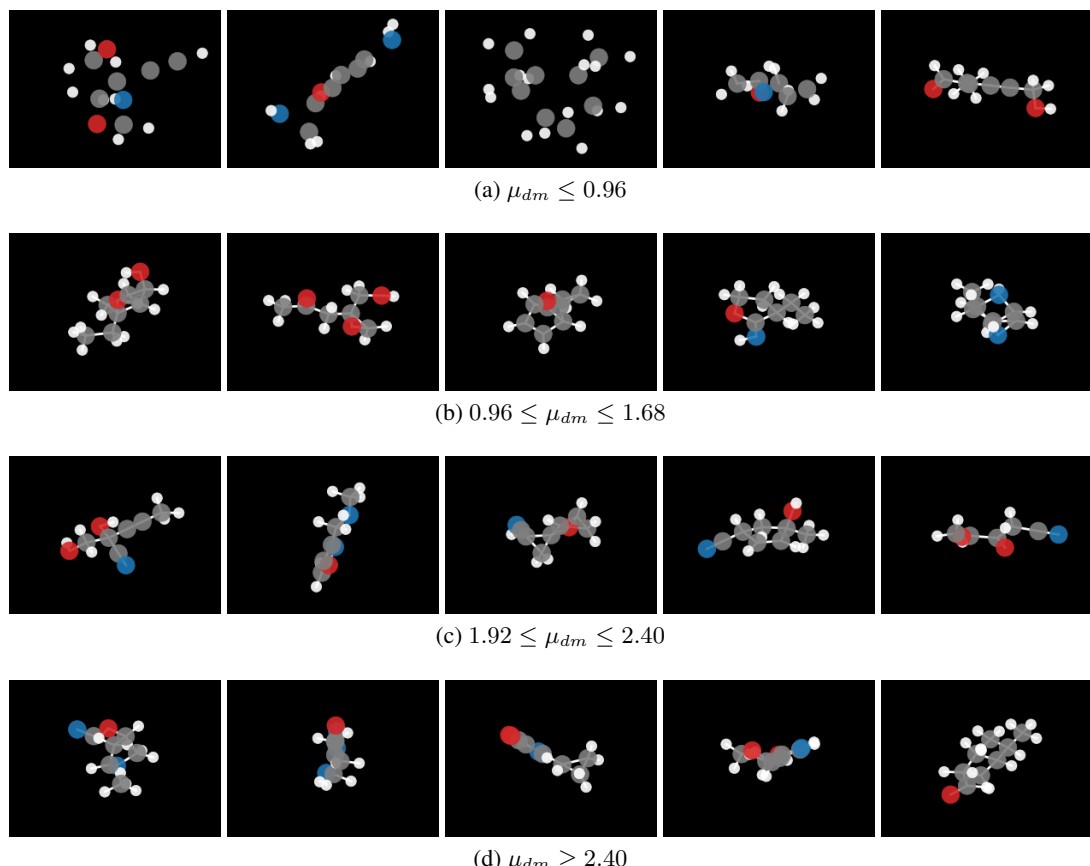

Figure 13: 3D GeoLDM+PRODIGY generations for Dipole Moment constraint for a range of values.

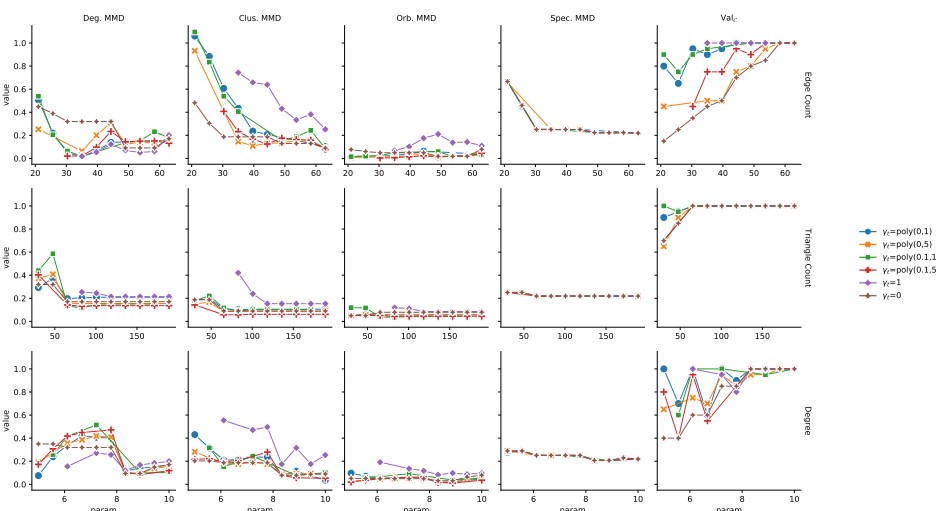

Figure 14: Value of different metrics for different $\gamma_t$ for Community Small on different constraints as we vary the parameters (param).

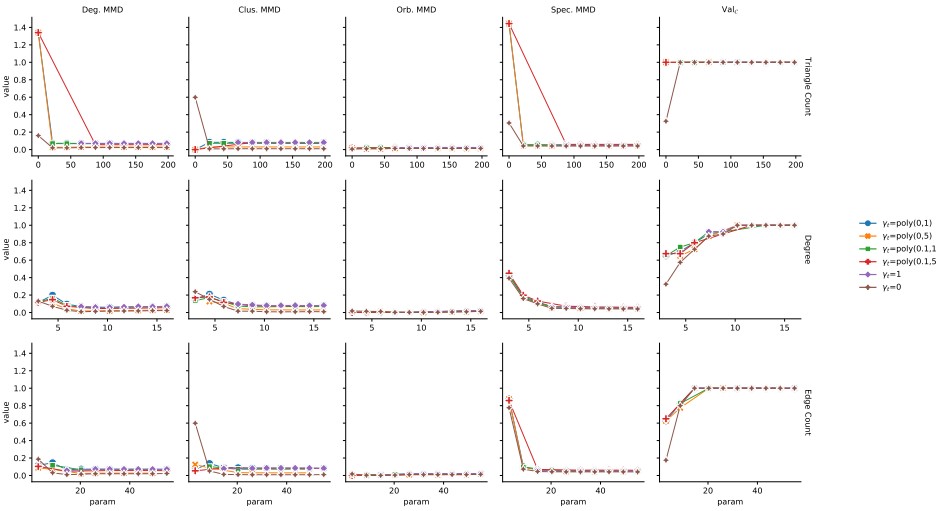

Figure 15: Value of different metrics for different $\gamma_t$ for Ego Small on different constraints as we vary the parameters (param).

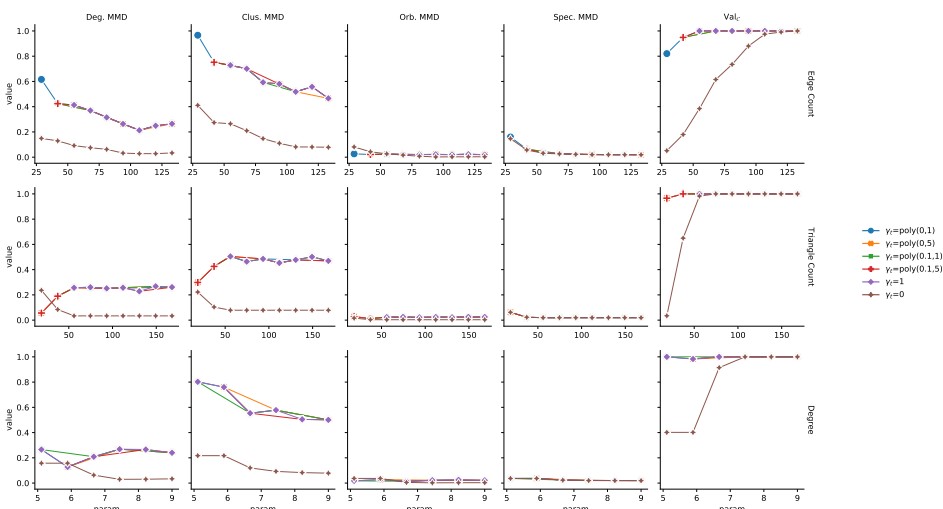

Figure 16: Value of different metrics for different $\gamma_t$ for Enzymes under different constraints as we vary the parameters (param).

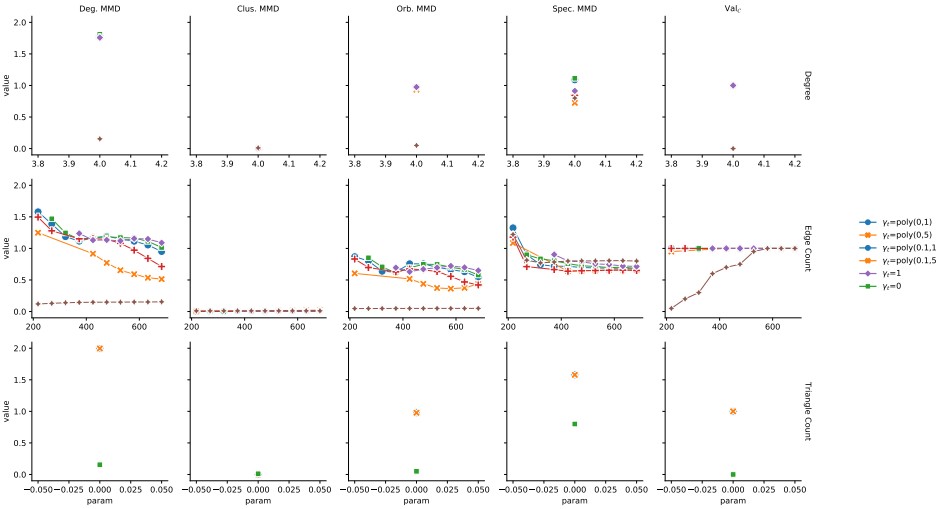

Figure 17: Value of different metrics for different $\gamma_t$ for Grid under different constraints as we vary the parameters (param).

