# OpenReview forum: "Plug-And-Play Controllable Graph Generation With Diffusion Models"
_ICLR.cc/2024/Conference — Submitted to ICLR 2024_

### Official Review · Reviewer_CFiy · 2023-10-18

**Soundness:** 2 fair
**Presentation:** 2 fair
**Contribution:** 2 fair
**Rating:** 6
**Confidence:** 3

**Summary:**

This paper proposes to have a constraint instead of a condition for generative modeling. The main evaluation metric is VAL_c, which measures the proportion of generated graphs that satisfy the constraint.

**Strengths:**

– constraint-based generation is a useful and novel avenue on graph generation

**Weaknesses:**

– It seems that the constrained generation settings is a completely different task from conditional generation (Hoogeboom 2022, Xu 2020) so, I think that the “constrained” keyword is more suitable?

– Since the tasks of Hoogeboom 2022, Xu 2020 are different, the comparisons with baselines are not really fair from what I read in the paper. You should compare the proposed “constrained” approach, with other constrained methods, maybe an adaptation of Bar-tal 2023 that you refer to? I am not convinced that measuring VAL_c on GDSS is fair against your model? This is why GDSS and EDP-GNN have the VAL_c metric approximately at zero, see Table 3.

**Questions:**

– Can your approach be applied also into conditional generation, e.g. Hoogeboom 2022 etc?

– “influences the sampling process in an obscure and uninterpretable manner”: can you elaborate?

– “Such controls are not differentiable and no method exists that can control the generation for these properties without relying on curating additional labeled datasets or retaining the entire generative mode”: from what I understand conditional generation is a different setup, so does not seem fair to compare what data is required in constrained generation ?

– Condition-based Control definition: Can you write what y and c are? Or link to the part where you formalize it.

– Constraint-based Control definition: can you clarify better the difference with soft control. Or refer to where you are defining it.

– Does the plug-and-play approach require training of 2 models? Could you point to where you discuss the advantage to training the soft constrained methods?

– Figure 2: Sampling process of PRODIGY (red) versus existing methods (Jo et al., 2022). Why are you comparing them, if in the experiments you are not?

---

> ### Author Response · Authors · 2023-11-19
> **Response to the Reviewer CFiy (1/2)**
>
> We thank the reviewer for providing constructive feedback on our work. Below, we provide detailed responses to their concerns and hope that the following discussion can encourage them to increase their score towards acceptance.
>
> > It seems that the constrained generation settings is a completely different task from conditional generation (Hoogeboom 2022, Xu 2020) so, I think that the “constrained” keyword is more suitable?
>
> We are not sure why the reviewer thinks we are focusing on conditional generation as we have never used that term anywhere to describe our approach. We would appreciate it if the reviewer could point out what has confused them. We do call our approach controllable generation as a constraint is a form of control.
>
> > Can your approach be applied also into conditional generation, e.g. Hoogeboom 2022 etc?
>
> Yes, as shown in Section 5.3 (dipole moment was also considered in Hoogeboom et al., 2022) and Appendix D.1, the problem of conditional generation can be solved using our approach. One can argue that the problem of conditional generation can be reduced to the more general constrained generation (the focus of this work). The objective of conditional generation can be seen as minimizing the difference (MAE or MSE) in the property of the generated samples and that of the given conditioning samples. More formally, this implies constraining the conditioned property of the generated samples within a certain bound ($\epsilon$) of the median/mean of the given conditioning samples. Theoretically, this will minimize the MAE/MSE of the property between the generated samples and the conditioning samples.
>
> > Since the tasks of Hoogeboom 2022, Xu 2020 are different, the comparisons with baselines are not really fair from what I read in the paper. You should compare the proposed “constrained” approach, with other constrained methods, maybe an adaptation of Bar-tal 2023 that you refer to? I am not convinced that measuring VAL_c on GDSS is fair against your model? This is why GDSS and EDP-GNN have the VAL_c metric approximately at zero, see Table 3
>
> >“Such controls are not differentiable and no method exists that can control the generation for these properties without relying on curating additional labeled datasets or retaining the entire generative mode”: from what I understand conditional generation is a different setup, so does not seem fair to compare what data is required in constrained generation ?
>
> We argue that comparison against base and conditional-generation models is fair in the absence of any constrained graph generation baseline in the literature.
>
> **Comparison with the base models:** To the best of our knowledge, there do not exist any baselines for constrained graph generation, that we can compare against. Thus, we decided to highlight the plug-and-play improvement over the base model. We test if our sampling strategy can improve the satisfaction of constraints that are not already satisfied by the original sampling methods. Our comparison with the base models in terms of $\text{Val}_C$ is to show that our method can improve the constraint validity significantly without significantly affecting the distributional metrics. Furthermore, we cannot have Bar-Tal et al. 2023 as a baseline since the proposed method in their work focuses on image generation and is limited to specific image-based constraints (panorama, aspect ratio, and spatial guiding) and any extensions of their method to graph-level constraints are not directly meaningful and hence not the focus of this work.
>
> **Comparison with conditional-generation models:** Our comparisons against the conditional-generation approaches are geared towards demonstrating that our approach is general enough that it can handle condition-based control as well. These experiments are not meant to assess the basic efficacy of our approach. We formulate the objective of their paper as a constraint and apply our method to achieve similar performance. In particular, we consider a constraint such that the value of the conditioned property is close (within $\epsilon$) to the mean/median of the given data. This would imply that the generated samples have property values closer to the conditioned value. In Section 5.3 and Appendix D.1, we thus compare the conditional generation methods with the constrained generation approach of PRODIGY. We use simple linear models to estimate these properties and find that even with these models, we can achieve lower MAE of the predicted properties between the generated and conditioning data samples. This implies that even with highly biased predictors, we can lower the error rate by effectively satisfying the constraint satisfaction and thus, keeping the property within $\epsilon$ bounds of the predicted median.

---

> ### Author Response · Authors · 2023-11-19
> **Response to the Reviewer CFiy (2/2)**
>
> > “influences the sampling process in an obscure and uninterpretable manner”: can you elaborate?
>
> An interpretable control on the generation should imply that the generated samples satisfy the given control. But conditional generation techniques instead consider the specified given control as a random variable and approximate the probability distribution $p(x | c(x) = c_0)$ where one can have $p(x=x’ | c(x) = c_0) > 0$ even though $c(x’) \neq c_0$. This is unexpected and each step in the sampling process thus becomes obscure and uninterpretable. Let us know if this doesn’t clarify the concern and we would be happy to elaborate.
>
> > Condition-based Control definition: Can you write what y and c are? Or link to the part where you formalize it.
>
> We note that $c(G, y)$ is a boolean function which is $1$ when $y_c(G) = y$ and $0$ otherwise. Here, $y_c(G)$ calculates some property of the graph $G$ and $y$ is a value that this property must take. As such, this is a standard definition in the existing literature and we discuss it in point 1 of controlled generation on page 3. We have also revised the definition in the updated draft.
>
> > Constraint-based Control definition: can you clarify better the difference with soft control. Or refer to where you are defining it.
>
> As we discuss in the Controlled Generation part of Section 2, the main difference is that in a soft control, we need to sample a graph $G$ given the control $c(G)$ as a dependent random variable, while for a hard control or constraint, we require the sampled graph $G$ to satisfy the control $c(G)$. For the soft control, one estimates a conditional probability distribution $p(G | c(G))$ and samples the graph $G$ from this. But this does not imply that the sampled graph would hold $c(G)$. This is because of the cyclic nature of the corresponding Bayesian network since a random variable $c(G)$ affects the distribution of $G$ (i.e., $p(G | c(G))$) and a given $G$ also affects the value of $c(G)$ (i.e., $p(c(G) | G)$). It thus forms an infinite Markov chain to necessitate that the sampled $G$ would imply $c(G)$. Constraint-based control specification, on the other hand, necessitates the control to be satisfied (hence, hard). We have clarified this better in the revised draft.
>
>
> > Does the plug-and-play approach require training of 2 models? Could you point to where you discuss the advantage to training the soft-constrained methods?
>
> We think the reviewer has misunderstood our approach regarding the necessity of training. We wish to clarify that we do not require training of two models. In fact, we do not train any new model but rather solve an optimization problem at every sampling step. The key contribution is to show that such an optimization problem can be solved efficiently, thus, removing the requirement of training a new predictor model or a new generative model, while allowing for the precise controls to be satisfied exactly. As we discuss in the Controlled Generation part of Section 2, the advantage over existing soft-constrained methods is two-fold – (1) Our method can be plugged onto any pre-trained diffusion model’s sampling without training any new model, and (2) We can satisfy precisely defined hard constraints on the generated samples that cannot be done using guidance-based methods.
>
> > Figure 2: Sampling process of PRODIGY (red) versus existing methods (Jo et al., 2022). Why are you comparing them, if in the experiments you are not?
>
> We are indeed comparing these two in the experiments. In tables 2,3,5,6, we compare PRODIGY (red) against GDSS (Jo et al., 2022).

---

> ### Author Response · Authors · 2023-11-21
> **Requesting feedback on the rebuttal**
>
> Thank you once again for your review! We have tried to carefully address your concerns in our responses and it would be very valuable to us if you could provide your feedback. If any issues still remain that need to be resolved, we would love to address them during the discussion period.

---

### Official Review · Reviewer_DbHE · 2023-10-18

**Soundness:** 3 good
**Presentation:** 2 fair
**Contribution:** 3 good
**Rating:** 6
**Confidence:** 3

**Summary:**

The authors propose a plug-and-play method to control the diffusion-based graph generation under certain constraints. In each step of diffusion, the generated graph is edited with the nearest feasible solution under the constraints, which can be solved with the Lagrangian method. The authors derive closed-form solutions for some well specified constraints commonly seen. Experiments show the proposed approach can effectively control the generation process to obtain results that satisfy the constraints.

**Strengths:**

1. Precise control with arbitrary constraints is a desired property for graph generation. The authors propose a general solution for certain classes of constraints. The authors derive closed-form solution for many commonly used constraints under the proposed projection approach. The detailed derivation may also inspire future research where new constraints may appear.
2. The empirical results are comprehensive. The authors show the proposed method can effectively control the generation direction towards graphs that satisfy the constraints, on both 2D graphs and 3D graphs. They further demonstrate the sensitivity of the approach to the interpolation value $\gamma_t$$. Analysis on the efficiency of the approach is also provided.

**Weaknesses:**

1. Though when the constraints cover all the graphs in the test set, the generated distribution is largely unaffected, in scenarios where the constraints do take effect, the generated distribution obviously deviates from the original distribution. And the deviation is model-sensitive. For example, in Table 2, with edge count as the constraint, the performance of GDSS downgrades by two folds on Clus. while EDP-GNN is largely unaffected. Also, on the dataset of Enzymes and Grid, both GDSS and EDP-GNN suffer a lot from the constraints. Furthermore, Figure 12 shows a lot of unrealistic molecular graphs (e.g. very large rings, disconnected graphs, very long bridging bonds) with the constraints on atom count.
2. The proposed projection paradigm main only applied to constraints with simple calculation process so that the optimization problem has a closed-form solution. When the constraints become complicated (e.g. involves non-linearity), it might be arduous or impossible to find such closed-form solution.

**Questions:**

1. Maybe a typo in section 4 right under equation 3: $\Pi_c(z) = \arg\min_{z\in c}||z-x||^2$，should be $\Pi_c(x) = \arg\min_{z\in c}||z-x||^2$

---

> ### Author Response · Authors · 2023-11-19
> **Response to the Reviewer DbHE**
>
> We are delighted by the reviewer’s positive recommendation of our work and we hope that any remaining concerns get addressed in the following discussion to further increase their scores. We also thank them for pointing out the typo that we have fixed in the revised version.
>
> > Though when the constraints cover all the graphs in the test set, the generated distribution is largely unaffected, in scenarios where the constraints do take effect, the generated distribution obviously deviates from the original distribution. And the deviation is model-sensitive.
>
> We would like to note that our performance difference in different base models is not a weakness but rather a strength of our method as we show plug-and-play improvement over these pre-trained models without being specific to any of them. Thus, as more advanced diffusion models are designed in the future, our method will automatically provide further improvement over them for constrained generation.
>
> > For example, in Table 2, with edge count as the constraint, the performance of GDSS downgrades by two folds on Clus. while EDP-GNN is largely unaffected. Also, on the dataset of Enzymes and Grid, both GDSS and EDP-GNN suffer a lot from the constraints.
>
> Please refer to our general comment on higher MMDs ([link](https://openreview.net/forum?id=xh0XzueyCJ&noteId=BJRokXMqZa)).
>
> > Furthermore, Figure 12 shows a lot of unrealistic molecular graphs (e.g. very large rings, disconnected graphs, very long bridging bonds) with the constraints on atom count.
>
> Unrealistic substructures such as large-cycle rings can arise as the constraint does not control for the size of the cycles and only cares about the counts of atoms. Furthermore, the probability distribution approximated by GDSS is not perfect and the original GDSS also samples such unrealistic graphs. However, the flexibility of our method allows us to control such unrealistic structures if needed by formalizing them as constraints. For example, large cycles can be controlled (just like we control the number of triangles). At this point, since finding the maximum cycle length in a graph is an NP-complete problem, controlling the maximum cycle length becomes a non-scalable problem for larger graphs (since we can’t even calculate it). Future works can explore other efficient methods to incorporate this constraint for larger graphs with certain approximations. Similarly, one can control for connectivity and bridging in the graph since both are formal graph-theoretic functions but we leave them for future works to consider.
>
> > The proposed projection paradigm main only applied to constraints with simple calculation process so that the optimization problem has a closed-form solution. When the constraints become complicated (e.g. involves non-linearity), it might be arduous or impossible to find such closed-form solution.
>
> We would like to note that this is the first work on constrained graph generation and to the best of our knowledge, PRODIGY is the only method that can support plug-and-play constrained generation of any graph-level constraint, once it’s formally defined.
> We consider a wide range of constraints and find that they have efficient projection operators by following the fixed template of Section 4.1. While we make certain assumptions about the form of the constraint, this does not restrict the applicability of this method to any new constraint. Contrary to the reviewer’s comment, we do not require a closed-form solution to exist for the optimization problem. In fact, as noted in Table 1 and Appendix A, the exact value of $\mu$ is never known in closed form and found using inexact linear and bisection search algorithms. Furthermore, we also consider approximations instead of exact projections for efficiency reasons and find superior results (see Triangle count constraint, Appendix A.2). All in all, while the proposed method requires solving an optimization problem, it does not require exact solutions and leave enough room for approximate and intricate solutions to be devised for specific constraints following the proof template (using Lagrangian operators) provided in Section 4.1 and Appendix A. We also discuss extensions to general non-linear constraints (assuming ReLU non-linearity) in Appendix B.3 but find that the optimization problem can get harder to solve and future works can focus on devising more efficient solutions.

---

> ### Author Response · Authors · 2023-11-21
> **Requesting feedback on the rebuttal**
>
> Thank you once again for your review! We have tried to carefully address your concerns in our responses and it would be very valuable to us if you could provide your feedback. If any issues still remain that need to be resolved, we would love to address them during the discussion period.

---

> > ### Comment · Reviewer_DbHE · 2023-11-22
> > **Thank you for the response**
> >
> > Thank you for the detailed response. By mentioning "closed-form solution", I mean closed-form solution for the projection operator in equation (4), that is, $\varphi_{\mu}^X$ and $\varphi_{\mu}^A$ has a analytical form (either with or without parameter $\mu$). For complicated functions, such analytical form might be arduous to derive (e.g. commonly used molecular property functions like QED and logP. Anyway, I admit this is a starting work in constrained graph generation, and I would like to keep my original positive score.

---

> > > ### Author Response · Authors · 2023-11-22
> > > **Thank you and further clarification**
> > >
> > > We really appreciate that the reviewer recognizes the novelty of our work. As the reviewer points out, complex molecular properties such as logP, QED, boiling points, etc. may not be known as a function of its structure (except for its wave function) and hence may lack a closed form for the projection operator. However, it is crucial to note that this is not a limitation of our method but rather a dependence on chemistry research that aims to find such closed-form functions of all molecular properties in terms of their structure. One way that is increasingly adopted by computational chemists is to learn neural networks from data for these properties (Hoogeboom et al., 2022). This gives us an exact predictive function of these properties and one can then constrain the predicted value from this parameterized model. This would imply constraining the value of the underlying property that we also validate through experiments (Appendix D.1 and Section 5.3 with simple parameterized linear models). We hope that our generic framework  inspires future works to study specialized constrained generation approaches for a variety of properties specific to application domains.

---

### Official Review · Reviewer_pKPg · 2023-10-30

**Soundness:** 2 fair
**Presentation:** 3 good
**Contribution:** 3 good
**Rating:** 6
**Confidence:** 4

**Summary:**

This work presents a conditional graph generation framework that can be applied in a plug-and-play manner to pretrained diffusion models. In particular, this work proposes to project the denoised samples to the constraint set at each sampling step which results in graphs that satisfy constraints, which could be applied to hard non-differentiable constraints that previous diffusion models are not applicable to.

**Strengths:**

- The paper is well-written and easy to follow, with sufficient background, related works, and problem setup.

- The motivation of this work, i.e., generating graphs from the data distribution that satisfies hard constraints, is clear which previous diffusion models are not directly applicable due to the non-differentiable constraints.

- This work presents various practical constraints on graph structures or properties (e.g., edge count, degree, valency, dipole moment) and the corresponding projection operators.

- The experimental results show that the proposed method is able to generate graphs that satisfy given conditions in a plug-and-play manner without re-training or fine-tuning the pre-trained diffusion models.

**Weaknesses:**

- The main concern is that using the proposed approach seems to have inferior generation quality compared to the original diffusion model. For example, in Table 2, GDSS+PRODIGY results in a significantly higher clustering coefficient MMD for the Community-small dataset and Enzymes dataset. This is problematic as the generated graphs should primarily follow the data distribution, not only satisfying the constraint.

- For some constraints, e.g., Degree and Molecular weight, the proposed method does not seem to achieve high validity (lower than 70%) even though the denoised samples are projected to the constrained set. In particular, GDSS+PRODIGY shows similar validity to GDSS for molecular weight constraint. Through analysis clarifying the reason for failing to satisfy the constraint is required.

I would like to raise my score if these concerns are sufficiently addressed.

**Questions:**

- Are the MMD results of Table 2 measured between the generated graphs and the constraint-filtered test set? I presume the MMD is not measured between generated graphs and the original test set as the MMD results are different among the constraints for the same dataset.

- What is the reason for GDSS+PRODIGY showing similar validity to GDSS for molecular weight constraint?

---

> ### Author Response · Authors · 2023-11-19
> **Response to the Reviewer pKPg**
>
> We appreciate the reviewer’s recognition of the strengths of our work and we hope the following discussion will be able to address their remaining concerns.
>
> > The main concern is that using the proposed approach seems to have inferior generation quality compared to the original diffusion model. For example, in Table 2, GDSS+PRODIGY results in a significantly higher clustering coefficient MMD for the Community-small dataset and Enzymes dataset. This is problematic as the generated graphs should primarily follow the data distribution, not only satisfying the constraint.
>
> Please refer to our general comment on higher MMDs ([link](https://openreview.net/forum?id=xh0XzueyCJ&noteId=BJRokXMqZa)).
>
> > For some constraints, e.g., Degree and Molecular weight, the proposed method does not seem to achieve high validity (lower than 70%) even though the denoised samples are projected to the constrained set. In particular, GDSS+PRODIGY shows similar validity to GDSS for molecular weight constraint. Through analysis clarifying the reason for failing to satisfy the constraint is required.
>
> > What is the reason for GDSS+PRODIGY showing similar validity to GDSS for molecular weight constraint?
>
> Lower satisfaction of given constraints in the experiments cited by the reviewer is primarily because we have to round our constraint-satisfying continuous solution to a corresponding discrete solution. As mentioned in Section 4, we consider a continuous relaxation of the constrained set, and the projection step projects the adjacency and atom-type matrices to the closest continuous solution that satisfies the constraint. At this point, we can guarantee the constraint is fully satisfied. However, since the graph structure is discrete, we round this continuous solution at the end to the closest discrete value and find the constraint validity on the rounded solution. This means that the constraint may not always be satisfied by the discrete solution. While specific rounding strategies to satisfy the constraints can be devised, this is a challenging task and must be carefully designed based on the application. For simplicity, we stick to the existing threshold-based rounding strategy as in the literature (Niu et al., 2020; Jo et al., 2022). In the experiments, we find that this is often very effective while being considerably efficient.
>
> Reflecting on the reviewer’s suggestion, we discuss and analyze possible reasons why low satisfaction may be achieved in specific cases including the ones cited by the reviewer:
>
> 1. Molecular weight constraint: This tends to be the case because the closest point found by the projection operator tends to make the $X$ matrix uniform across different atom types. When a low satisfaction is achieved on this constraint, we see in the failure cases that it tends to converge to $X_{ij} \rightarrow W / (n \sum_j m_j)$. This is not ideal during rounding as even though $\sum_{i, j} m_j X_{ij} \le W$, we can have $\sum_{i} m_{\arg\max_j X_{ij}} > W$ since $X_{ij} \approx X_{ik}$ ($j \ne k$). This can be alleviated with additional constraints on $X$ (such as $|X_{ij} - X_{ik}| > \epsilon$ for some $\epsilon$) but we leave such engineering tweaks for specific constraints for future works to examine.
> 2. Ego-small (Edge Count and Degree): We believe this comes due to the limited number of discrete graphs that are possible around the constraint-feasible set. Ego-small consists of typically sparse graphs with few (~5) nodes. Our constraints of at most $3$ edges and at most $5$ maximum degree allow for very few graphs that tend to be harder to reach through simple rounding techniques.
> 3. EDP-GNN, Community-small (Edge Count and Degree): We believe the low satisfaction, when it happens in this case, comes because we consider an aggressive $\gamma_t = 1$ projection for EDP-GNN. This may not be ideal as it can drift the sampling procedure too much leading into a specific continuous region that may not be close to the learned distributions of (discrete) graphs. EDP-GNN only consists of Langevin sampling and thus, one cannot schedule a $\gamma(t)$ as a function of $t$ as in GDSS. But we believe a scalar value of $\gamma_t < 1$ might give better results and we will update the results in the paper to include this (will share here as soon as they are available).
>
> > Are the MMD results of Table 2 measured between the generated graphs and the constraint-filtered test set? I presume the MMD is not measured between generated graphs and the original test set as the MMD results are different among the constraints for the same dataset.
>
> Yes, as mentioned in Section 5.2, MMD results in Table 2 are found using the generated and constraint-filtered test graphs. Since in this experiment, we are only interested in generating graphs that satisfy specific constraints, unsatisfiable graphs do not form meaningful ground-truth samples for comparison. Hence, we filtered these out to compute the MMD.

---

> > ### Comment · Reviewer_pKPg · 2023-11-21
> > **Thank you for the response**
> >
> > Thank you for the detailed response, especially the analysis of possible reasons why low satisfaction may be achieved.
> >
> > Although I think the problem this work is targeting (i.e., constrained graph generation) to be important and the proposed method to be novel in the context of graph generation, I think that the experiments could be further improved.
> >
> > - Since the authors claim that MMD w.r.t. some properties cannot completely identify a graph distribution, there should be alternative metrics (or other experiments) to show that the samples from the PRODIGY do belong to the data distribution. Current results fail to demonstrate this.
> >
> > - I do not understand the t-test results in the general comments. Which MMD (Deg., Clus., Orb.) was the target for the t-test? I find the Clus. results on Enzymes and Deg. results on Community-small to be significantly different on GDSS.

---

> > > ### Author Response · Authors · 2023-11-21
> > > **Response to the follow-up question 1/2**
> > >
> > > We thank the reviewer for their prompt response to our rebuttal. We provide further clarifications to their follow-up questions regarding the experimental results below and hope this can encourage them to revisit their score.
> > > > Since the authors claim that MMD w.r.t. some properties cannot completely identify a graph distribution, there should be alternative metrics (or other experiments) to show that the samples from the PRODIGY do belong to the data distribution. Current results fail to demonstrate this.
> > >
> > > We believe that the reviewer has misunderstood our comment about MMD not completely identifying a distribution. This comment was to point out that it is not ideal to compare the raw MMD numbers as it does not have an inherent scale (as pointed out in O’Bray et al., 2021). Hence, it is not meaningful to compare raw MMDs in solitude but rather see it in light of other objectives (e.g. constraints in our case) that one is interested in. To motivate this, we then give an example that even though the MMDs seem to be low for GDSS (baseline model), none of the generated grid graphs by that baseline model are valid grids. One can then argue that samples from GDSS original model also do not belong to the data distribution but evaluating graph distributions is not the focus of this work. Since our focus is to provide a plug-and-play approach to constrained graph generation (which as the reviewer points is a new and important problem), we stick to the existing metrics that have been shown to work well for these pre-trained graph diffusion models (i.e., EDP-GNN (Niu et al., 2020) and GDSS (Jo et al., 2022)).
> > >
> > > Furthermore, we agree with the reviewer that it is important to show that samples from PRODIGY belong to the data distribution (i.e., preserve data characteristics to be more precise). To this end, we would like to clarify two points.
> > >
> > >   1. In Table 2, we are dealing with a highly constrained setting, which falls under the high-uncertainty (low accuracy) regions of the learned probability distribution of the base models. Thus, it is not an ideal setting to test the hypothesis that PRODIGY matches the data distribution (however, we still obtain comparable MMD numbers). Instead, to demonstrate that PRODIGY preserves the data distribution, we consider a more suitable setting when the constraint objective overlaps with the probability distribution learned through the unconditional generation objective (see Section 5.4, Tables 5, 6). Here, one can see that the samples from PRODIGY closely match the data distribution.
> > >   2. While we rely on MMD for the lack of better metrics for general graphs, we do use other metrics (valency validity, novelty, Frechet Chemical distance, atomic stability, molecular stability) for molecular datasets. In Tables 3 and 4, we show that we preserve these properties of molecules while satisfying the constraints.
> > >
> > > It would be very helpful if the reviewer could elaborate any of their outstanding concerns if they still remain or cite an example experiment that they feel is missing.
> > >
> > > **References**
> > >
> > > Niu, Chenhao, et al. “Permutation invariant graph generation via score-based generative modeling”. AISTATS 2020.
> > >
> > > Jo, Jaehyeong, et al. “Score-based Generative Modeling of Graphs via the System of Stochastic Differential Equations”. ICML 2022.
> > >
> > > O'Bray, Leslie, et al. "Evaluation metrics for graph generative models: Problems, pitfalls, and practical solutions." ICLR 2022.

---

> ### Author Response · Authors · 2023-11-21
> **Response to the follow-up question 2/2**
>
> > I do not understand the t-test results in the general comments. Which MMD (Deg., Clus., Orb.) was the target for the t-test? I find the Clus. results on Enzymes and Deg. results on Community-small to be significantly different on GDSS.
>
> In the general comment, we consider the hypothesis that given a constraint, the MMD of GDSS/EDP-GNN for any property is different from the MMD of GDSS+PRODIGY/EDP-GNN+PRODIGY for that property and dataset. Thus, we combine different properties in the same statistical analysis because each property is as important as the other and find no statistically significant result for the same.
>
> To specifically address what the reviewer has suggested, one must instead test the hypothesis that x-MMD(GDSS/EDP-GNN | dataset) is different than x-MMD(GDSS/EDP-GNN+PRODIGY | dataset), where x is Deg., Clus., Orb.. We give the results for these tests below. We highlight the rows which are found to be statistically different ($p <0.05$) and note that a positive t-statistic mean x-MMD(GDSS | dataset) > x-MMD(GDSS+PRODIGY | dataset) while a negative t-statistic mean x-MMD(GDSS/EDP-GNN | dataset) < x-MMD(GDSS/EDP-GNN+PRODIGY | dataset). This shows that PRODIGY statistically worsens the MMD in only the deg. and orb. MMDs for Grid (that we already discussed above). This is given the fact that PRODIGY significantly lowered the MMDs in 7 cases and improved the constraint validity in all cases. This establishes that the strengths of PRODIGY in satisfying constraints overshadow increasing the MMD in one dataset (Grid).
>
> | Model | Dataset | MMD | t-statistic | p-value |
> | -- | -- | -- | -- | -- |
> | EDP-GNN | Community-small | **Deg.** | **4.43** | **0.01** |
> | | | Clus. | -2.67 | 0.06 |
> | | | Orb. | -0.97 | 0.38 |
> | | | Combined | -0.52 | 0.63 |
> | | Ego-small | Deg. | -0.70 | 0.52 |
> | | | **Clus.** | **3.65** | **0.02** |
> | | | **Orb.** | **3.60** | **0.02** |
> | | | Combined | 1.70 | 0.16 |
> | | Enzymes | Deg. | -1.45 | 0.22 |
> | | | **Clus.** | **44.0** | **1.59e-06** |
> | | | Orb. | -0.18 | 0.86 |
> | | | Combined | -0.02 | 0.98 |
> | | Grid | **Deg.** | **-9.82** | **6.02e-4** |
> | | | **Clus.** | **inf** | **0.0** |
> | | | **Orb.** | **-15.56** | **9.94e-05**|
> | | | Combined | -0.69 | 0.52 |
> | GDSS | Community-small | Deg. | 0.50 | 0.64 |
> | | | Clus. | -0.97 | 0.38 |
> | | | Orb. | 0.22 | 0.83 |
> | | | Combined | -0.25 | 0.82 |
> | | Ego-small | Deg. | -0.88 | 0.43 |
> | | | **Clus.** | **3.10** | **0.04** |
> | | | Orb. | 0.95 | 0.39 |
> | | | Combined | 1.32 | 0.26 |
> | | Enzymes | Deg. | -0.79 | 0.47 |
> | | | Clus. | -1.92 | 0.13 |
> | | | Orb. | 1.06 | 0.35 |
> | | | Combined | -0.96 | 0.39 |
> | | Grid | **Deg.** | **-6.91** | **2.29e-3** |
> | | | **Clus.** | **15.50** | **1.01e-4** |
> | | | **Orb.** | **-6.48** | **2.91e-3** |
> | | | Combined | -1.79 | 0.15 |

---

> > ### Comment · Reviewer_pKPg · 2023-11-23
> > **Thank you for detailed response**
> >
> > Thank you for the response. The t-test results have addressed my concerns about MMD results of PRODIGY. I understand that the experiments of Table 2 deal with highly constrained settings and MMD is not a perfect metric for measuring if the generated samples are really from the data distribution. Therefore, quantitatively showing if the generated graphs preserve the data characteristics, for example, satisfying the two community structures for Community-small or the chain-like structure of Enzymes would further strengthen the authors' claim. In this sense, evaluation on benchmark datasets used in SPECTRE, DiGress, or DruM (e.g., Planar dataset or SBM dataset) is highly recommended.
> >
> > I raise my score from 5 to 6 as the authors have addressed my concerns, but still evaluate this work as borderline.

---

> > > ### Author Response · Authors · 2023-11-23
> > > **Thank you for increasing the score**
> > >
> > > We thank the reviewer so much for their efforts in assessing our work and for revisiting their score. We absolutely agree that the evaluation of other benchmark datasets would further strengthen our work. In fact, we are highly interested in providing results on DruM as an underlying model since our method can be directly plugged into it and DruM shows high validity on the Planar and SBM datasets. We have already reached out to DruM authors to share their code and pre-trained models but note that DruM is a contemporaneous work currently [under review at ICLR 2024](https://openreview.net/forum?id=UQVhOVhUi4) and hence the models are not yet available. In the absence of the DruM models, we decided to omit the comparison with GDSS as it gives a validity of 0 on the datasets cited by authors. To clarify, this validity pertains to the valid graphs generated by the underlying unconditional model, not to be confused with the constraint validity computed after applying our approach.
> > >
> > > To this end, we promise to add the full spectrum of results on DruM as soon as the authors make their code or trained models available to us.

---

### Official Review · Reviewer_86tq · 2023-11-01

**Soundness:** 2 fair
**Presentation:** 3 good
**Contribution:** 3 good
**Rating:** 5
**Confidence:** 5

**Summary:**

This paper introduces PRODIGY (PROjected DIffusion for Generating constrained Graphs), a plug-and-play methodology for generating graphs that adhere to designated constraints by leveraging pre-trained diffusion models. Addressing the intricacies of controllable graph generation, the technique integrates a projection operator into the reverse diffusion process to ensure alignment with the specified constrained space. Notably, PRODIGY augments the capabilities of existing diffusion models to satisfy stringent constraints without compromising the proximity to the original data distribution.

**Strengths:**

The proposed method implements controllable graph generation by applying a plug-and-play approach based on the pre-trained diffusion model without fine-tuning, this is a new perspective to reduce experimental costs.

**Weaknesses:**

1. My biggest concern is whether such an operation, which pulls the embedding towards the designated constrained space through projection during the reverse process, is effective in more practical scenarios. The paper proposes some compromises to determine whether to pull towards the original data distribution or the constrained space during the reverse process. However, from the experimental results, I find that the performance of this approach is not entirely satisfactory. For example, in the Community-small and Ego-small datasets, the MMD metric worsens in many cases after applying the PRODIGY method. Additionally, in the QM9 molecular generation experiment, adding PRODIGY poses a significant risk of decreasing validity and novelty.

2. Another concern I have pertains to the potential impact of such operations on the convergence of the Langevin Dynamics process. It would be prudent to provide a proof for the convergence of the Projected Inexact Langevin Dynamic/Algorithm follows some ideas from PSLA [1].

3. I also have some concerns regarding the practicality of the proposed method. From the projection operations summarized in the paper, the proposed method seems more suitable for controllable generation for atomic features X or structural information A related properties, such as valency, atom count, and molecular weight. However, in real-world applications, such as constrained molecular generation, we mostly expect the generated molecules to exhibit certain graph-level properties.

Minor:

4. Some notations should be introduced when they are first proposed, such as $\textbf{Z}_\theta$ in Eq. (3).

[1] Lamperski, A. (2021, July). Projected stochastic gradient langevin algorithms for constrained sampling and non-convex learning. In Conference on Learning Theory (pp. 2891-2937). PMLR.

Updated:
The enhancement achieved with PRODIGY seems to be marginal. It's possible that a detailed convergence analysis might reveal the limited performance improvement.

**Questions:**

Why does the method in the paper only demonstrate its effectiveness on continuous diffusion methods and not on recently proposed discrete diffusion methods, such as DiGress+PRODIGY?

---

> ### Author Response · Authors · 2023-11-19
> **Thank you for your review**
>
> We thank the reviewer for recognizing the strengths of our work and providing constructive feedback to improve it further. Below, we respond to specific concerns raised by the reviewer:
>
> > My biggest concern is whether such an operation, which pulls the embedding towards the designated constrained space through projection during the reverse process, is effective in more practical scenarios. The paper proposes some compromises to determine whether to pull towards the original data distribution or the constrained space during the reverse process. However, from the experimental results, I find that the performance of this approach is not entirely satisfactory. For example, in the Community-small and Ego-small datasets, the MMD metric worsens in many cases after applying the PRODIGY method. Additionally, in the QM9 molecular generation experiment, adding PRODIGY poses a significant risk of decreasing validity and novelty.
>
> Please refer to our general comment on higher MMDs ([link](https://openreview.net/forum?id=xh0XzueyCJ&noteId=BJRokXMqZa)) in some dataset/model settings on general graphs. In addition, the reviewer raised concerns regarding a decrease in validity in QM9, which happens for EDP-GNN under molecular weight and atom count constraints. We would like to note that (1) This is attributed to the underlying model of EDP-GNN as it only denoises the adjacency matrix while these constraints are dependent on only the node attribute matrix (thus, the reverse sampling of $A$ is not affected by the projection of $X$ and can decrease the validity), and (2) Our framework gives the ability to further increase the validity directly by using the valency constraint on top of any other constraint.
> Further, decreased novelty under atom count constraint in QM9 primarily arises from the fact that we study a highly constrained setting, where we need to generate molecules with only C and O atoms as compared to the unconstrained setting where we can generate over C, N, O, and F. This naturally limits the total number of possible molecules and thus, limits the novel molecules we can generate over the C and O molecules that already exist in the dataset. Thanks to the reviewer’s comments, we have realized that comparing raw novelty metrics is unfair to our method since we are exploring a limited set of molecules compared to GDSS. We believe a better metric to study would be the proportion of molecules that are both novel and constraint-valid. For this example, we can see it would be roughly $0.33 \times 81.04 = 26.74\%$ for GDSS while it would be $1.00 \times 67.67 = 67.67\%$ for GDSS+PRODIGY. We will add the constraint-valid and novelty (VN%) metric in the revised version.

---

> ### Author Response · Authors · 2023-11-19
> **Theoretical convergence**
>
> > Another concern I have pertains to the potential impact of such operations on the convergence of the Langevin Dynamics process. It would be prudent to provide a proof for the convergence of the Projected Inexact Langevin Dynamic/Algorithm follows some ideas from PSLA [1].
>
> We appreciate this suggestion but note that such a proof of convergence is extremely non-trivial and out of the scope of this work.
>
> Having said that, as we note in Section 2 (Projected Sampling), modern diffusion models (Song et al., 2021; Jo et al., 2022) differ from theoretical Langevin dynamics considered in Lamperski, 2021, and others. This implies that the proofs in PSLA cannot be directly transferred to our current work due to various reasons –
> 1. We do not know of a corresponding convergence proof even for the unconditional generation of diffusion-based models. Since our method is based on these models, any proof of convergence of the overall sampling becomes highly non-trivial.
> In the theoretical works, the (unnormalized) probability distribution is known exactly and assumed to have a certain form (uniformly sub-Gaussian), while such assumptions do not hold for diffusion models where the probability density function (or equivalently, the score function) is approximated through arbitrary (graph) data. For real-world graph data, such sub-Gaussian assumptions are not well-motivated.
> 2. We consider a non-zero drift coefficient (which is assumed to be $0$ in PSLA) for more expressivity, which complicates the theoretical analysis further.
> 3. Our proposed method is a generalization of PSLA in the sense that we do not project the noisy sample completely but instead move the sample only slightly in the direction of the projection (see the last paragraph of Section 4). In particular, we consider $G_{t-1} = (1 - \gamma_t)\tilde{G_{t-1}} + \gamma_t \prod_{C} \tilde{G_{t-1}}$, where $\tilde{G_{t-1}}$ is obtained from $G_t$ following the reverse process. In the experiments, we find that $\gamma_t = (t/T)^p $ for some $p$ gives the best results, which is in contrast with PSLA which is an instantiation of our method with $\gamma_t = 1$.
>
> Further, reflecting upon the reviewer’s comments, we analyzed the drift caused by the PRODIGY method on the original sampling by bounding $\lVert G_T - \tilde{G_T} \rVert$ assuming the same initial random graph $G_0$ and Reverse$(G_t) \sim$ Reverse$(\tilde{G_t})$. In particular, we show that $\lVert G_T - \tilde{G_T} \rVert\le 2 \sum_{t} \gamma_t \lVert \prod_{C}(G_t) - \tilde{G_t} \rVert$. Since the term inside the summation is what we optimize in every iteration under a given constraint, our method optimizes the upper bound to be minimal. Furthermore, $\gamma_t$ gives the flexibility to be close to the original graphs if needed. To prove the statement, we first note that $\lVert G_T - \tilde{G_T} \rVert = \lVert (G_T - G_{T-1}) + (G_{T-1} - G_{T-2}) + \cdots + (G_1 - G_0) - (\tilde{G_T} - \tilde{G_{T-1}}) - (\tilde{G_{T-1}} - \tilde{G_{T-2}}) - \cdots - (\tilde{G_1} - G_0) \rVert$ $\le \lVert (G_T - G_{T-1}) - (\tilde{G_T} - \tilde{G_{T-1}}) \rVert + \cdots + \lVert (G_1 - G_0) - (\tilde{G_1} - G_0) \rVert$. By definition, $G_t - \tilde{G_T} = \gamma_t (\prod_{C}(G_t) - \tilde{G_t})$. Thus, we get $\lVert G_T - \tilde{G_T} \rVert\le 2 \sum_{t} \gamma_t \lVert \prod_{C}(G_t) - \tilde{G_t} \rVert$.
> Finally, we note that a lack of proof does not undermine the effectiveness of our work since we show empirically that PRODIGY can (1) converge to the underlying model’s learned distribution when the test set is a subset of the constraint feasible set (see Section 5.4), and (2) generate graphs that satisfy hard constraints (see Section 5.2).

---

> ### Author Response · Authors · 2023-11-19
> **Practicality of the method**
>
> > I also have some concerns regarding the practicality of the proposed method. From the projection operations summarized in the paper, the proposed method seems more suitable for controllable generation for atomic features X or structural information A related properties, such as valency, atom count, and molecular weight. However, in real-world applications, such as constrained molecular generation, we mostly expect the generated molecules to exhibit certain graph-level properties.
>
> We would like to clarify that our method is suitable for and indeed applies to graph-level properties. Any (global) property $h$ of an attributed 2-D graph $G = (X, A)$ can be written as a function of its parts $h(G) = h(X, A)$. We consider general constraints of the form $(h_1(X, A) \le 0) \land (h_2(X, A) \le 0) \land \cdots (h_k(X, A) \le 0)$. For instance, the number of triangles and edges are all global properties of the graph, while total molecular weight and number of certain atoms are global properties of a 2D molecule. We also consider the dipole moment, which is a global property of the 3D molecular structure. Finally, we also show extensions to general linear (Appendix B.2, D.1) and non-linear properties (with ReLU non-linear activations in Appendix B.3) of a graph using parameterized models. In this work, we show, for the first time, that one can control generation from pre-trained models under precisely defined constraints efficiently on a wide array of graph-level properties. Future works can extend our approach to satisfy specific constraints more efficiently.
>
> One can further argue that the exact form of complex molecular properties as a function of its constituents may not be known when these are only experimentally observed (for example, boiling points, chemical reactivity, etc.). However, this is not a limitation of our method but rather a dependence on chemistry research that aims to find such closed-form functions of all molecular properties in terms of their structure. One way that is increasingly adopted by computational chemists is to learn neural networks from data for these properties (Hoogeboom et al., 2022). This gives us an exact predictive function of these properties and one can then constrain the predicted value from this parameterized model. This would imply constraining the value of the underlying property that we also validate through experiments (Appendix D.1 and Section 5.3 with simple parameterized linear models).
>
> We hope this helps clarify the concern but if the reviewer still believes some graph-level properties would not be allowed in our proposed framework, then it would be great if they could give specific examples of the same.

---

> ### Author Response · Authors · 2023-11-19
> **Discrete diffusion models**
>
> > Why does the method in the paper only demonstrate its effectiveness on continuous diffusion methods and not on recently proposed discrete diffusion methods, such as DiGress+PRODIGY?
>
> The focus of our work on supporting plug-and-play sampling-based constrained graph generation for continuous-time diffusion models is a design choice borne out of both the recently reported generation results of state-of-the-art methods in this field and a carefully conducted feasibility analysis on our end.
>
> First, it is not clear that discrete models are necessarily the best choice and provide state-of-the-art performance on unconditional graph generation. As a case in point, in Section 2 (start of page 3), we note that more recent advancements in continuous-time models (DruM, Jo et al., 2023) have been shown to outperform discrete models such as DiGress (however, note that the codes and models for DruM are not publicly available yet but we are in communication with their authors and our method can directly apply on their trained models without any modifications).  As for the DiGress model cited by the reviewer, we recognize that their ability to do conditional generation (separate focus from ours but their definition of conditional generation is generalizable to our constraint-based formulation) is useful and provides a meaningful comparison point for us. Considering this, in Appendix D.1, we compared the guidance-based approach of DiGress against our constrained generation result with GDSS to generate molecules with given HOMO and dipole moment values. We found PRODIGY to provide comparable performance even with weaker underlying continuous time models. This is because we can generate molecules close to the median property value which reduces the error even with a highly biased model.
>
> In terms of feasibility, the discrete models pose significant efficiency challenges for constrained generation. These challenges primarily stem from the combinatorial explosion of the space of constraint-feasible graphs. The constraint set for discrete graphs is formed by a set of discrete points with no definite structure, unlike the continuity and convexity in the continuous case. Due to these reasons, the solution to the optimization problem becomes an instance of integer programming, and thus, results in an NP-hard problem. The projection step would then involve iterating over all the graphs in the constrained set to find the closest one. This explodes as the size of the constrained set increases, which can be exponential even for a simple edge-count constraint since $|E| \le B$ has $\binom{n^2}{B}$ different graphs that satisfy the constraint (where n is the number of nodes). Thus, constraint satisfaction for discrete diffusion models can be NP-hard and we leave it for future works to explore innovative workaround solutions.
>
> Finally, we would like to note that this is the first work on constrained graph generation, and to the best of our knowledge, PRODIGY is the only method that can support plug-and-play constrained generation of any graph-level constraint, once it’s formally defined. We believe that it is an unfair expectation for a single work to solve a highly novel and challenging problem in all available settings.
> In light of this, we would highly appreciate it if you would be kind to reassess our work and score it based on the above clarifications. We are happy to engage in further discussions and would love to provide any clarifications that can help alleviate any of your outstanding concerns.
>
> *Jo, Jaehyeong, Dongki Kim, and Sung Ju Hwang. "Graph Generation with Destination-Driven Diffusion Mixture." arXiv preprint arXiv:2302.03596 (2023).*

---

> ### Author Response · Authors · 2023-11-21
> **Requesting feedback on the rebuttal**
>
> Thank you once again for your review! We have tried to carefully address your concerns in our responses and it would be very valuable to us if you could provide your feedback. If any issues still remain that need to be resolved, we would love to address them during the discussion period.

---

> ### Comment · Reviewer_86tq · 2023-11-23
>
> Thank you for your response. Could you please revise your current version and mark the revisions as blue?

---

> > ### Comment · Reviewer_86tq · 2023-12-04
> >
> > I have observed the updated revision of your work. However, in my view, the enhancement achieved with PRODIGY seems to be marginal.

---

### Author Response · Authors · 2023-11-19
**Addressing general concerns**

We appreciate the time and efforts of all the reviewers in providing constructive feedback on our study. We are delighted by their recognition of our work's novelty in constrained graph generation and appreciation for the plug-and-play nature of the proposed solution approach in satisfying hard constraints. We've carefully addressed the queries and concerns of the reviewers in specific responses below and revised our manuscript accordingly.

Reviewers, particularly 86tq, pKPg, and DbHE, expressed concerns about the high MMD scores in Table 2. We respond to this below.

MMD measures if two sample sets come from different distributions by comparing expectations over specific sample functions (Gretton et al., 2012). However, it's important to note that high MMD scores in some cases don't necessarily diminish our method's effectiveness for several reasons:
- Firstly, MMD with respect to some properties like degree, clustering coefficient, and orbit count cannot completely identify a graph distribution. This is highlighted in the relatively low MMDs of GDSS (baseline model) on the Grid dataset even though it does not generate a single valid grid graph (since grid graphs always have a maximum degree of 4 and zero triangles). This is one of the examples where PRODIGY outshines significantly as it increases the constraint validity from 0 to 100% as shown in Table 2.
- Secondly, the constrained graphs studied in Table 2 form a large minority (in fact, maximally minority) of the dataset and thus, likely belong to the high uncertainty regions of the trained diffusion models. This means satisfying these constraints can result in exploration in regions where the diffusion models do not approximate the probabilities well. Thus, high MMDs can be a result of poorly learned underlying distribution and not necessarily our sampling method. We establish in Table 5 that when the constraint set significantly overlaps with the dataset (and thus, belongs to the low uncertainty region of the model), then we can indeed achieve low MMDs, even surpassing the base models.
- Reflecting on reviewers’ comments, we further explore the impact of PRODIGY sampling on MMDs, by conducting two different analytical experiments:
    1.  We performed a 2-sample t-test to see if PRODIGY's MMDs were statistically different from the base model. The results, showing p-values > 0.05, suggest no significant difference. We provide the corresponding result across all datasets below:
| Constraint | Model | t-statistic | p-value |
| -- | -- | -- | -- |
| Edge-Count | EDP-GNN | -0.12 | 0.91 |
| | GDSS | -1.48 | 0.15 |
| Triangle Count | EDP-GNN | -1.37 | 0.18 |
| | GDSS | -1.38 | 0.18 |
| Degree | EDP-GNN | -0.61 | 0.55 |
| | GDSS | -1.63 | 0.12 |
    2. We adjusted our constraints to align better with the test set by additionally bounding it with a trivial lower bound of the minimum test property and explored a more gradual $\gamma_t$ growth, following Appendix D.4 analysis. The outcomes in Table 10 and Appendix D.4 (page 19) indicate that our method can achieve low MMDs while meeting constraints.

Gretton, Arthur, et al. "A kernel two-sample test." The Journal of Machine Learning Research 13.1 (2012): 723-773.

---

### Author Response · Authors · 2023-11-23
**Summary of the Reviews**

We extend our gratitude to the review panel for their diligent efforts and insightful feedback on our manuscript. This communication aims to summarize the reviewers' evaluations and our subsequent responses during the rebuttal period.

The reviewers have acknowledged the innovation of our study, focusing on constrained graph generation and the efficacy of our plug-and-play methodology for controlled generation without the necessity of retraining. Key highlights include:
  - Recognition of the **novelty and significance of constrained graph generation**, as noted by reviewers pKPg, DbHE, and CFiy.
  - Commendation for the **clarity and quality of our manuscript** by reviewers pKPg and DbHE.
  - Appreciation of our **plug-and-play approach for controlled generation without retraining**, highlighted by reviewers 86tq and pKPg.
  - Acknowledgment of our exploration of **various practical constraints** by reviewers pKPg and DbHE.
  - Positive remarks on our **comprehensive experimental results** and the inclusion of a sensitivity analysis, particularly noted by reviewer DbHE.

However, the reviewers have expressed concerns regarding specific aspects of our work, including concerns regarding the high MMDs in Table 2, incapability to possibly constrain complex properties, and other clarification questions, which we have diligently addressed as follows:

  - To effectively address the queries raised by reviewers 86tq, pKPg, and DbHE regarding the elevated MMD scores for some case documented in Table 2, we provided a detailed response in the section [“Addressing general concerns”](https://openreview.net/forum?id=xh0XzueyCJ&noteId=BJRokXMqZa), employing statistical tests to demonstrate that the **differences in MMD scores are statistically insignificant compared to the baseline**. We also outline the limitations in studying raw MMD numbers, especially for the constrained setting. These are acknowledged by the reviewer pKPg with an increase in score from 5 to 6.
  - In response to concerns raised by reviewers 86tq and DbHE, we reaffirmed that our framework, as delineated in Section 3.1, **can accommodate any graph-level property**, provided it is formally defined. We also referenced our use of parameterized approximations in Appendix D.1 and Section 5.3 for instances where exact definitions are unfeasible.
  - Reviewer pKPg raised issues regarding the complete infeasibility of constraints in certain scenarios. Our comprehensive analysis in this regard led to a positive reassessment by the reviewer, increasing their score from 5 to 6.
  - To address concerns by reviewer 86tq regarding convergence, we established a straightforward yet potent upper bound on the divergence from the original process under ideal conditions.
  - We provided additional clarifications regarding the comparison with conditional generation and base models (CFiy), and an extended discussion on discrete diffusion models (86tq).

In summary, our research studies a novel problem of generating graphs that satisfy precise constraints and introduces a pioneering plug-and-play sampling approach, PRODIGY. This method innovatively utilizes pre-trained diffusion models for graphs, enabling the generation of graphs from an underlying distribution that can meet any given constraint during sampling. Our work takes a significant stride in the field of graph generation, specifically offering both theoretical and practical advancements towards enabling interpretable control in the graph generative models across different domains.

---

### Meta-Review · Area_Chair_kQ1U · 2023-12-08

**Metareview:**

In this submission, the authors proposed a method called PRODIGY, applying projected diffusion to generate graphs with constraints. The proposed method actually provides an algorithmic framework compatible with any pre-trained diffusion models, allowing them to sample reasonable graphs under specific constraints. Experiments demonstrate the rationality of the proposed method to some degree.

Strengths: (a) The topic itself is important --- making graph generation processes controllable is valuable for many applications, especially in drug discovery. (b) The proposed method is a plug-and-play module, compatible with many existing methods.

Weaknesses: (a) More than two reviewers have concerns about the rationality of the technical route and the solidness of the experimental part. Although the authors made efforts to resolve the concerns, providing more experimental results. This problem is not fully solved because 1) some representative baselines are missed, and 2) the improvements achieved by the proposed method are too incremental.

Overall, this submission has some merits indeed, but more analytic and experimental parts are required to enhance the rationality and effectiveness of the proposed method.

**Justification For Why Not Higher Score:**

Reviewers and AC have concerns about the effectiveness of the method. The improvements achieved by the method are too incremental.

**Justification For Why Not Lower Score:**

N/A

---

### Decision · Program_Chairs · 2024-01-16

Reject